# FLOW-MATCHING BASED REFINER FOR MOLECULAR CONFORMER GENERATION

## ABSTRACT

Low-energy molecular conformers generation (MCG) is a foundational yet challenging problem in drug discovery. Denoising-based methods include diffusion and flow-matching methods that learn mappings from a simple base distribution to the molecular conformer distribution. However, these approaches often suffer from error accumulation during sampling, especially in the low SNR steps, which are hard to train. To address these challenges, we propose a flow-matching refiner for the MCG task. The proposed method initializes sampling from mixed-quality outputs produced by upstream denoising models and reschedules the noise scale to bypass the low-SNR phase, thereby improving sample quality. On the GEOM-QM9 and GEOM-Drugs benchmark datasets, the generator–refiner pipeline improves quality with fewer total denoising steps while preserving diversity.

## 1 INTRODUCTION

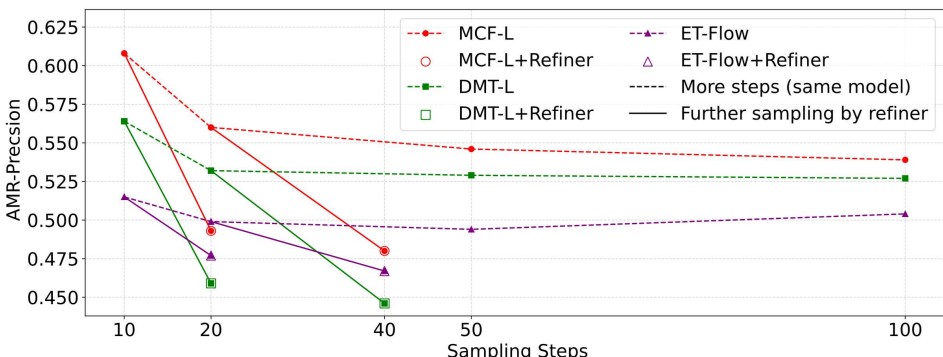

Figure 1: GEOM-DRUGS: increasing sampling steps vs. adding a refiner. Metric: Average Minimum RMSD (AMR)↓. The x-axis shows *total* sampling steps (including refiner steps). We compare allocating extra budget to adding our refiner (8.3M) against simply increasing steps for three strong baselines: MCF-L (242M), ET-Flow (8.3M), and DMT-L (150M). For ET-Flow, the base architecture and predictor–corrector (PC) schedule are identical; only the refiner is added and fine-tuned.

Low-energy 3D conformations, called *conformers*, determine a molecule's biological, chemical, and physical properties. (Guimaraes et al., 2012; Schütt et al., 2018; Klicpera et al., 2019; Axelrod & Gomez-Bombarelli, 2020; Schütt et al., 2021) Therefore, generating accurate and diverse ensembles of conformers from the molecular graph is a fundamental task in computational chemistry. Traditional approaches can be grouped into two main categories: physics-based methods, such as molecular dynamics(Pracht et al., 2020), which explore conformational space with high fidelity at high computational cost; and cheminformatics methods (Hawkins et al., 2010; Riniker & Landrum, 2015), which are more efficient but less accurate, often trading precision for speed.

Denoising-based generative models, including diffusion (Ho et al., 2020; Song et al., 2021) and flow matching (Lipman et al., 2023; Albergo & Vanden-Eijnden, 2023; Liu et al., 2023; Tong et al., 2024),

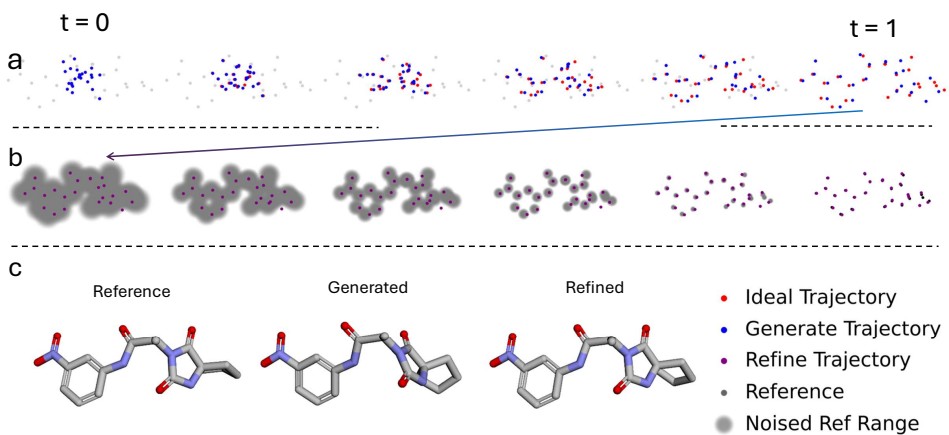

Figure 2: Overview of the proposed Refiner pipeline. Figs. (a) and (b) show PCA projections of the 3D atomic coordinates. (a) A diffusion/flow-matching generator maps a Gaussian noise prior to coarse 3D conformers, where geometric errors between the ideal and generated conformers gradually accumulate along the sampling trajectory. (b) During training, the Refiner only requires its inputs to cover the real-data geometry, rather than exactly match the test-time distribution. During sampling, it truncates the accumulated errors by rescaling the noise, thereby further correcting the upstream conformers. (c) Visualization on a representative molecule, where most errors in the upstream conformers are corrected by the Refiner, leading to higher-quality final structures.

have developed rapidly in recent years. Recent machine learning methods address this gap in the MCG task by learning to sample the distribution of low-energy conformers, aiming for both speed and accuracy. Existing work includes (1) methods based on pairwise atom distances in a distance matrix (Luo et al., 2021; Shi et al., 2021); however, they have too many degrees of freedom, which often leads to unstable optimization and poor results. (2) methods operate in torsional angle space (Jing et al., 2022); unfortunately, the performance is limited by the use of local information obtained by preprocessing tools such as RDKit. (3) More recent methods that predict 3D coordinates and report the current SOTA performance (Xu et al., 2022; Wang et al., 2024; Hassan et al., 2024; Liu et al., 2025). They work directly in 3D coordinate space to achieve the current best results, especially with large-scale models. However, the methods based on 3D atom coordinates still suffer issues from the denosing method: models are hard to train to predict the score or vector field of steps dominated by noise. Karras et al. (2022). Early diffusion steps that start from pure noise incur large errors. Because denoising is sequential and removes noise gradually, these early errors can accumulate and propagate through the trajectory, degrading final quality (Li & van der Schaar, 2024; 2023; Li et al., 2023; Chung et al., 2022).

We address this by coupling a standard denoising model with a refiner. Instead of beginning from pure noise, the refiner starts from an upstream generator's conformer: a plausible structure perturbed with error, which can be considered as some noise; rather than a noise-only state. By skipping the high-noise phase, early-step errors are substantially smaller than in traditional denoising, improving final performance.

Empirically, as shown in Fig. 1, simply adding steps improves only marginally, whereas our method achieves larger improvements with fewer additional steps; this even holds under a fixed model architecture with the same PC sampler.

Our contributions can be summarized as follows:

1. We introduce a denoising model plus *refiner* pipeline for molecular conformer generation and theoretically justify its design.

2. We identify the properties that make the refiner effective and validate them theoretically and empirically.

3. The pipeline reaches higher quality with fewer total sampling steps; this gain remains even when the refiner model structure is the same as the previous method.

4. The proposed refiner is plug-and-play across the conformer generated by different upstream models and needs no per-model tuning.

## 2 RELATED WORK

**Diffusion and Flow Matching.** Diffusion models (Song et al., 2021; Ho et al., 2020) generate high-quality, diverse samples by learning the score $\nabla_{\mathbf{x}} \log p_t(\mathbf{x})$ of noise-perturbed data and numerically integrating a reverse-time SDE/ODE from a Gaussian prior to the data distribution. Flow Matching (FM) (Lipman et al., 2023; Albergo & Vanden-Eijnden, 2023; Liu et al., 2023) instead regresses a target vector field defined by stochastic interpolants, which also enables training with arbitrary source distributions (Pooladian et al., 2023). Recent work unifies diffusion and FM under stochastic interpolants, revealing them as closely related denoising generative models. (Albergo et al., 2023; Ma et al., 2024; Yu et al., 2024). However, in very low SNR regimes where noise dominates, the training targets become hard to learn (Karras et al., 2022; Hang et al., 2024), and during the early denoising steps, errors can accumulate and propagate, degrading sample quality (Li & van der Schaar, 2024; Chung et al., 2022; Li & van der Schaar, 2023; Li et al., 2023),.

**Refiners.** Refinement modules are widely used in coarse-to-fine pipelines to recover high-frequency detail. Podell et al. (2023) introduces a diffusion-time–conditioned stage that polishes images, and Pandey et al. (2022) refines reconstructions produced by an upstream VAE. Restoration-style refiners target specific degradations such as deblurring and super-resolution (Whang et al., 2022; Saharia et al., 2022), However, and typically rely on a particular upstream model or error type. Other methods repair outputs by injecting noise and then denoising (Kawar et al., 2022; Sawata et al., 2022); however, such noise-injection–based refiners must still start sampling by revisiting high-noise steps thereby risking instability.

**Molecular Conformer Generation.** Deep learning methods for molecular conformer generation have attracted growing interest in recent years. Early attempts applied diffusion to distance matrices (Luo et al., 2021; Shi et al., 2021), but these approaches underperformed. Ganea et al. (2021) predict conformers via learned chemical structure parameters (e.g., bond lengths, bond angles, and torsions), yet overall quality is constrained by torsional accuracy; Jing et al. (2022) improves torsion prediction but remains limited by local structure. Subsequent works model and perturb directly in atomic coordinate space and scale to larger architectures (Xu et al., 2022; Wang et al., 2024; Hassan et al., 2024; Liu et al., 2025), achieving state-of-the-art performance. Nevertheless, sampling quality remains constrained by the intrinsic behavior of sequential denoising procedures.

## 3 PRELIMINARIES

**Flow Matching for 3D Molecular Conformers**: Following Hassan et al. (2024), Given a molecular graph $\mathcal{G}$ and all-atom coordinates $\mathbf{x} \in \mathbb{R}^{N \times 3}$, flow matching learns a time-dependent SE(3)-equivariant vector field $\mathbf{v}_\theta(\mathbf{x}, t, \mathcal{G})$ that transports a tractable base to the low-energy target.

**Interpolant and training.** Sample a coupling $(\mathbf{x}_0, \mathbf{x}_1) \sim \rho_0 \times \rho_1$ (conditioned on $\mathcal{G}$), and define a interpolant with stochastic

$$\mathbf{x}_t = \alpha(t)\,\mathbf{x}_0 + \beta(t)\,\mathbf{x}_1 + s(t)\,\mathbf{z}, \qquad t \in [0,1],\ \mathbf{z} \sim \mathcal{N}(\mathbf{0}, \mathbf{I}), \tag{1}$$

with $\alpha(0) = 1,\ \beta(0) = 0,\ \alpha(1) = 0,\ \beta(1) = 1$. Its instantaneous (non-parametric) velocity is

$$\mathbf{u}_t := \tfrac{\mathrm{d}}{\mathrm{d}t}\mathbf{x}_t = \alpha'(t)\,\mathbf{x}_0 + \beta'(t)\,\mathbf{x}_1 + s'(t)\,\mathbf{z}. \tag{2}$$

The optimal transport velocity is $\mathbf{v}_t^\star(\mathbf{x}) = \mathbb{E}[\mathbf{u}_t \mid \mathbf{x}_t = \mathbf{x}, t]$. Flow matching trains the model $\mathbf{v}_\theta$ via supervised regression:

$$\mathcal{L}_{\mathrm{FM}}(\theta) = \mathbb{E}\Big[\big\| \mathbf{v}_\theta(\mathbf{x}_t, t, G) - \mathbf{u}_t \big\|_2^2\Big], \tag{3}$$

The Base priors $\mathbf{x}_0$ is: a bonded harmonic prior is used to preserve local geometry, or pure Gaussian noise: $\mathbf{x}_0 = \sigma\,\varepsilon$. Therefore, when training and starting sampling at $t = 0$, there's no signal.

## 4 FLOW-MATCHING-BASED REFINER

**Motivation.** Current molecular conformer generation (MCG) methods typically rely on denoising generative models (diffusion or flow matching), which currently achieve state-of-the-art performance (Hassan et al., 2024; Liu et al., 2025; Wang et al., 2024) compared to alternative paradigms. However, early high-noise sampling steps are difficult to learn (Karras et al., 2022); when sampling starts from pure noise, the initial stage suffers from large errors. At the same time, denoising sampling is sequential; these stepwise errors will propagate and accumulate along the sampling process, and finally harm the final quality. (Li & van der Schaar, 2024; Chung et al., 2022; Li & van der Schaar, 2023; Li et al., 2023)

For the MCG task, we work in $\mathbf{x} \in \mathbb{R}^{3N}$ for a molecular conformer with $N$ atoms. Let $\mathbf{x}_1$ be the ground-truth coordinates and $\hat{\mathbf{x}}_1$ denotes the generated conformer. Then we can get RMSD $\Delta = \hat{\mathbf{x}}_1^{\star} - \mathbf{x}_1$ by Kabsch alignment (Kabsch, 1976) where $\hat{\mathbf{x}}_1^{\star}$ is the conformer after alignment. This error can be considered as a kind of noise $\Delta \sim \mathcal{N}(0, \sigma^{\star}\mathbf{I}_{3N})$ with an unknown real noise scale $\sigma^{\star}$.

This suggests a simple shift: instead of starting from pure noise, we initialize sampling from upstream-generated conformers $\hat{\mathbf{x}}_1^{\star}$, thereby skipping the inherently hard-to-learn high-noise phase. Moreover, flow matching (FM) admits an arbitrary choice of the base distribution for $\mathbf{x}_0$ (Pooladian et al., 2023). Doing so alleviates early-stage error propagation and yields a smaller initial sampling error; it finally improves overall stability. Accordingly, we propose a refiner that further polishes conformers generated by the prior approaches rather than regenerating them from scratch. Our design addresses three challenges:

(a) **Unknown scale.** With $\sigma^{\star}$ unknown at test time, how can we define an effective scale on the fly?

(b) **Scale realignment.** How should we set or adapt $\sigma$ to bypass the pure-noise phase and enter a well-trained regime?

(c) **Diversity preservation.** How can we improve quality without collapsing the diversity of upstream conformers?

### 4.1 REFINER DEFINITION

**Base distribution $x_0$ in training**. Since our goal is to *refine* upstream conformers rather than regenerate them, we depart from the usual pure-noise initialization and adopt a base distribution $\mathbf{x_0}$ that retains the ground-truth signal. Therefore, we define

$$\mathbf{x}_0 = \mathbf{x}_1 + \sigma\,\varepsilon, \qquad \varepsilon \sim \mathcal{N}(0, \mathbf{I}_{3N}). \tag{4}$$

This is permitted by flow matching, which allows an arbitrary choice (Pooladian et al., 2023). In implementation, we set $\sigma = 1$, which has the following properties:

(1) At $t = 0$, sampling does not start from pure noise but from a data-centered state that already contains signal, thereby skipping the pure-noise phase. Moreover, the scale $\sigma = 1$ is well below the variance-exploding (VE) regime (Song et al., 2021), which covers the data only under very large noise.

(2) The initial perturbation $\sigma\,\varepsilon$ exceeds the error level of prior methods. By introducing the Wilson–Hilferty approximation (Wilson & Hilferty, 1931), it gives: when $\sigma = 1$, for a conformer with $N = 10$ heavy atoms perturbed by Gaussian noise $\sigma\varepsilon$, the upper RMSD bound with 95% Confidence Interval is about 1.98 Å, which is notably larger than the typical errors of prior methods (details in Appendix A.1). Because the schedule scales the noise as $(1 - t)\sigma$, whose continuity in $t$ ensures value-range coverage of $t^{\star} \in (0, 1)$: there exists $t^{\star} \in (0, 1)$. Here, we denote $t^{\star} \in [0, 1]$ as the unknown effective time at which the refiner's noise scale matches that of the upstream conformer, i.e., $(1 - t^{\star})\sigma = \sigma^{\star}$. such that

$$(1 - t^{\star})\sigma = \sigma^{\star}, \qquad t^{\star} = 1 - \frac{\sigma^{\star}}{\sigma}. \tag{5}$$

This justifies a sampling-time self-calibration: once $(1 - t)\sigma \approx \sigma^{\star}$, the effective refinement begins at $t^{\star}$.

---

**Algorithm 1** REFINER (ODE)

| **Training** | **Sampling** |
|---|---|
| 1: **repeat** | **Require:** Generated sample $\hat{\mathbf{x}}$ and its graph $\mathcal{G}$ |
| 2:     **sample** $\mathbf{x}_1$, $\mathcal{G} = (\mathcal{V}, \mathcal{E}) \sim p_{\text{data}}$ |     Trained refiner model $\mathbf{u}_\theta$ |
|       $\varepsilon \sim \mathcal{N}(\mathbf{0}, I_{3N})$,    $t \sim \text{Uniform}(0,1)$ |     Number of steps $N$ |
| 3:     $\mathbf{x}_0 \leftarrow \mathbf{x}_1 + \sigma\varepsilon$ | 1: Schedule $\{t_n\}_{n=0}^N$ with $t_0 = 0, t_N = 1$ |
| 4:     $\mathbf{x}_0 \leftarrow \text{Align}_{\text{Kabsch}}(\mathbf{x}_0, \mathbf{x}_1)$ | 2: $\mathbf{x} \leftarrow \hat{\mathbf{x}}$ |
| 5:     $\mathbf{x}_t \leftarrow (1-t)\mathbf{x}_0 + t\mathbf{x}_1 + s(t)\mathbf{z}$ | 3: **for** $n = 0$ **to** $N-1$ **do** |
| 6:     $\mathbf{u}_t \leftarrow -\sigma\varepsilon + s'(t)\mathbf{z}$ | 4:     $\Delta t_n \leftarrow t_{n+1} - t_n$ |
| 7:     **Predict** $\hat{\mathbf{u}} \leftarrow \mathbf{u}_\theta(\mathbf{x}_t, t, \mathcal{G})$ | 5:     $\mathbf{x} \leftarrow \mathbf{x} + \Delta t_n \cdot \mathbf{u}_\theta(\mathbf{x}, t_n, \mathcal{G})$ |
| 8:     $\mathcal{L} \leftarrow \|\hat{\mathbf{u}} - \mathbf{u}_t\|_2^2$;   update $\theta \leftarrow \theta - \eta\nabla_\theta\mathcal{L}$ | 6: **end for** |
| 9: **until** convergence | 7: **return x** |

---

**Interpolant $\mathbf{x}_t$ and velocity $\mathbf{u}_t$ of the refiner.** Given $\mathbf{x}_1$ and a base distribution defined by 4. We use the linear interpolant:

$$\mathbf{x}_t = (1-t)\mathbf{x}_0 + t\mathbf{x}_1 + s(t)\mathbf{z}$$
$$= \mathbf{x}_1 + (1-t)\sigma\varepsilon + s(t)\mathbf{z}. \tag{6}$$

The corresponding velocity is

$$\mathbf{u}_t = \tfrac{\mathrm{d}}{\mathrm{d}t}\mathbf{x}_t = -\sigma\varepsilon + s'(t)\mathbf{z}. \tag{7}$$

Following Hassan et al. (2024), we schedule $s(t) = \sqrt{t(1-t)}$ to control the instantaneous velocity, which preserves value-range coverage of the noise scale:

$$s'(t) = \frac{1-2t}{2\sqrt{t(1-t)}}\mathbf{z}, \qquad \mathbf{u}_t = -\sigma\varepsilon + \frac{1-2t}{2\sqrt{t(1-t)}}\mathbf{z} \tag{8}$$

**Objective function** We train a time-conditioned vector field $\mathbf{u}_\theta(\cdot, t)$ to match the target velocity along the interpolant:

$$\min_\theta \mathbb{E}\left[\|\mathbf{u}_\theta(\mathbf{x}_t, t, \mathcal{G}) - \mathbf{u}_t\|_2^2\right], \qquad t \sim \text{Unif}[0,1] \tag{9}$$

**Sampling**, given an upstream sample, we set $\mathbf{x}_0 = \widetilde{\mathbf{x}}$ and sampling

$$\frac{\mathrm{d}}{\mathrm{d}t}\mathbf{x}_t = \mathbf{u}_\theta(\mathbf{x}_t, t), \qquad \mathbf{x}_{t=0} = \mathbf{x}_0, \tag{10}$$

to obtain the refined conformer $\mathbf{x}_{t=1}$. The Detailed can be found in Algorithm 1. We are also following the same correction strategy as Hassan et al. (2024)

**Design implications.** The proposed base distribution and $t$-schedule address challenges **(a)** and **(b)** as follows: for **(a)**, by ensuring a smaller noise in the value range coverage. The conformers by the upstream model with error no larger than $\sigma\varepsilon$ lie within the refiner's reachable range, enabling on-the-fly realignment without knowing $\sigma^\star$. For **(b)**, the base distribution in Eq. 4 is data-centered rather than pure noise, so sampling does not begin in the pure-noise regime and thus bypasses it.

Also, this design introduces a new challenge: **(d)** step time mismatch. Because effective refinement begins near $t^\star$ rather than at the real start of the sampling, the pair $(\mathbf{x}_t, t)$ may be distributionally mismatched, which can harm the quality. In the next section, we address challenges **(c)** and **(d)** by detailing the model's representations and update rules, which maintain diversity and are robust to the $(\mathbf{x}_t, t)$ mismatch.

## 4.2 PROPERTIES

Refiner behavior under a mismatch in time ($t^\star \neq t$), aiming to preserve diversity and ensure robustness so that already good states are not downgraded. Our analysis centers on how atom coordinates are represented by the dynamic part of the representation.

**Representation properties.** We first analyze the model's representation properties. Following prior denoising-based approaches (Xu et al., 2022; Jing et al., 2022; Hassan et al., 2024), we parameterize

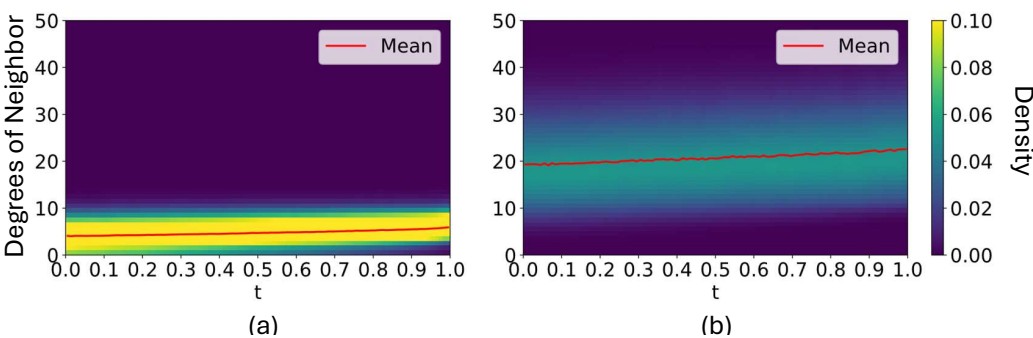

Figure 3: Comparison of neighbor degree distributions during training: (a) with a maximum radius of 2.5; (b) with a maximum radius of 5.0.

the refiner velocity model $\mathbf{u}_\theta(\mathbf{x}_t, t, \mathcal{G})$ as in Eq. 9. After removing components that are not dependent on $t$, including $\mathcal{G}$ or $\mathbf{x_1}$ (in Eq. 6, the $\mathbf{x_1}$ is part of $\mathbf{x_t}$), the remaining $t$-dependent representation can be written as $(1-t)\,\sigma\,\varepsilon\ +\ s(t)\,\mathbf{z}$. Consequently, the magnitude of the velocity target defined in Eq. 7 is positively correlated with it.

**Relative representation.** By SE(3)-equivariance, we may fix atom $i$ at the origin through a global rigid transform. The representation is the collection of relative vectors $r_{ij}(t)\ =\ \mathbf{x}_j(t) - \mathbf{x}_i(t)$. together with the neighbor set induced by a radius threshold $R$: $\mathcal{D}_R(i,t)\ =\ \big\{\,j \neq i \mid \|r_{ij}(t)\| \leq R\,\big\}$. After alignment at $t = 1$, denote the decomposition

$$r_{ij}(t)\ =\ r_{ij}(1)\ +\ \Delta r_{ij}(t), \tag{11}$$

where $r_{ij}(1)$ is the static reference part and $\Delta r_{ij}(t)$ collects all time-varying perturbations. Thus, the representation that may mismatch with $t$ includes: (i) distance perturbation: $\Delta d_{ij}(t)$; (ii) angular deviation: $\overrightarrow{r_{ij}}(t)'$, (iii) neighbor degree under radius $R$: $\mathcal{D}_R(i,t)$.

**Representation properties when $t$ mismatch..** We focus on scenarios with a time mismatch $t^\star > t$, where $(1 - t^\star)\sigma < (1 - t)\sigma$ and thus $\sigma^\star = (1 - t^\star)\sigma < \sigma$. We analyze the consequences of this mismatch for the position vector $r_{ij}$ and the neighbor degree $\mathcal{D}_R(i,t)$ as follows:

Firstly, we approximate the time-mismatch effect on a pair by an origin-centered Gaussian perturbation:

$$r_{ij}(t)\ =\ r_{ij}(1)\ +\ \sigma_{\text{eff}}\,\varepsilon, \qquad \varepsilon \sim \mathcal{N}\big(0, I_3\big), \tag{12}$$

with $\sigma_{\text{eff}} = \sqrt{2}$ when each endpoint has isotropic noise at both endpoints; equivalently, after fixing $x_i$ at the origin, the effective perturbation on $r_{ij}$ has variance $\sigma_i^2 + \sigma_j^2$, which is equal to $2\sigma^2$ under identical scales.

Then we can get (i) when distance & direction under time mismatch: Both the distance perturbation $\Delta d(t)$ and the angular deviation of $r_{ij}(t)'$ relative to $r_{ij}$ are positively correlated with the noise scale $\sigma_{\text{eff}}$ (Anderson et al., 1958; Mardia & Jupp, 2009), therefore, configurations attainable under a smaller scale $\sigma^\star$ are contained within those under a larger scale $\sigma$ .

(ii) when the neighbor degree under time mismatch: The neighborhood size $|\mathcal{D}(i)|$ (with a fixed radius cutoff $R$) directly scales the magnitude of aggregated vector features. Because neighbors at different distances encode distinct representations, we empirically quantify the degree by randomly subsampling under a fixed $R$ and counting $|\mathcal{D}(i)|$ (see Fig. 3). As $t$ increases during training while the effective noise $(1-t)\sigma$ decreases, $|\mathcal{D}(i)|$ monotonically increases. When test states correspond to $t^\star > t$, higher-degree configurations may be underrepresented relative to earlier-$t$ training samples, which can induce errors at the beginning of refinement.

**Sampling dynamic properties when $t$ mismatch.** We summarize properties relevant to the mismatch between $(\mathbf{x}_t, t)$ and $(\mathbf{x}_{t^\star}, t^\star)$, assuming that noise drives the dynamics and correlates with the target.

1. Distance-dominant scaling under SE(3). With an SE(3)-equivariant backbone where orientation is not modeled explicitly, direction provides a limited independent signal. Update magnitudes scale

Table 1: Performance of generated conformer ensembles on the GEOM-DRUGS test set, reported as Coverage (COV, %) and Average Minimum RMSD (AMR, Å). Coverage (COV) is computed with threshold $\delta = 0.75$ Å. The best performance is **bold**.

| Method | Recall | | | | Precision | | | |
|---|---|---|---|---|---|---|---|---|
| | COV $\uparrow$ | | AMR $\downarrow$ | | COV $\uparrow$ | | AMR $\downarrow$ | |
| | Mean | Med | Mean | Med | Mean | Med | Mean | Med |
| GeoMol | 44.60 | 41.40 | 0.875 | 0.834 | 43.00 | 36.40 | 0.928 | 0.841 |
| GeoDiff | 42.10 | 37.80 | 0.835 | 0.809 | 24.90 | 14.50 | 1.136 | 1.090 |
| Torsional Diff. | 72.70 | 80.00 | 0.582 | 0.565 | 55.20 | 56.90 | 0.778 | 0.729 |
| MCF-L (242M) | 85.10 | 92.86 | 0.390 | 0.343 | 66.63 | 70.00 | 0.623 | 0.546 |
| ET-Flow (8.3M) | 80.15 | 85.71 | 0.458 | 0.429 | 73.89 | 80.56 | 0.556 | 0.494 |
| DMT-L (150M) | 85.95 | 91.98 | 0.378 | 0.353 | 67.97 | 71.97 | 0.599 | 0.529 |
| MCF-L + **Refiner** (8.3M) | 86.44 | 93.68 | 0.368 | 0.330 | 72.07 | 78.41 | 0.550 | 0.480 |
| ET-Flow + **Refiner** | 80.29 | 85.11 | 0.439 | 0.410 | 74.58 | **81.59** | 0.530 | 0.467 |
| DMT-L + **Refiner** | **87.47** | **94.12** | **0.349** | **0.319** | **75.91** | 81.51 | **0.497** | **0.446** |
| Boost (%) | 1.95 | 1.97 | 7.44 | 8.36 | 2.73 | 1.28 | 12.03 | 10.94 |

Table 2: Performance of generated conformer ensembles on the GEOM-QM9 test set, reported as Coverage (COV, %) and Average Minimum RMSD (AMR, Å). Since recent work already achieves 100% median COV at the commonly used threshold $\delta = 0.5$ Å and a median AMR below 0.05 Å, we adopt the more challenging COV threshold of $\delta = \mathbf{0.05}$ Å. The best performance is **bold**.

| Method | Recall | | | | Precision | | | |
|---|---|---|---|---|---|---|---|---|
| | COV (**0.05**) $\uparrow$ | | AMR $\downarrow$ | | COV (**0.05**) $\uparrow$ | | AMR $\downarrow$ | |
| | Mean | Med | Mean | Med | Mean | Med | Mean | Med |
| GeoDiff | - | - | 0.297 | 0.229 | - | - | 1.524 | 0.510 |
| GeoMol | - | - | 0.225 | 0.193 | - | - | 0.270 | 0.241 |
| Torsional Diff. | - | - | 0.178 | 0.147 | - | - | 0.221 | 0.195 |
| MCF-B (62M) | 66.82 | 67.86 | 0.101 | 0.050 | 61.18 | 64.29 | 0.117 | 0.059 |
| ET-Flow (8.3M) | 75.72 | 87.23 | 0.083 | 0.031 | 70.32 | 75.00 | 0.114 | 0.053 |
| DMT (55M) | 72.90 | 83.33 | 0.087 | 0.036 | 67.76 | 75.00 | 0.107 | 0.047 |
| MCF-B + **Refiner** (8.3M) | 74.87 | 81.82 | 0.100 | 0.035 | 77.41 | 94.44 | 0.101 | 0.023 |
| ET-Flow + **Refiner** | 78.40 | 88.89 | 0.076 | 0.028 | 77.36 | 89.94 | 0.103 | 0.031 |
| DMT-B + **Refiner** | **79.50** | **89.44** | **0.070** | **0.026** | **80.37** | **97.92** | **0.076** | **0.021** |
| Boost (%) | 3.86 | 2.22 | 11.22 | 14.60 | 13.72 | 22.78 | 35.77 | 55.32 |

primarily with distance perturbations, which are strongly correlated with the effective noise scale $\sigma_{\text{eff}}$. Consequently, cleaner inputs with larger $t^\star$ induce smaller predicted velocities $\|\mathbf{u}_t\|$.

2. Degree mismatch co-occurs with low velocity. Neighbor degree mismatch is most likely at higher noise. Mismatch is more likely to happen when the conformers generated by the upstream model are cleaner than the refiner's current state, i.e., $t^\star > t$. Under this condition, the velocity will be relative low as well.

**Representation mismatch implications** Taken together, the dynamic properties of our representation directly address the above challenges: **(c)** If a generated conformer is already near a target basin, the predicted velocities are low-magnitude, which ensures a small update and avoids basin switching, therefore preserving diversity. **(d)** mismatch arises when the refiner's current time step is smaller, i.e., $t^\star > t$. This makes a warm-up phase in which some of the atoms' neighborhood degrees may be out of distribution, but the associated velocities are small, so errors won't be catastrophic.

## 5 EXPERIMENTAL RESULT

We evaluate the generator–refiner pipeline and, via controlled studies, isolate the refiner's contribution to empirically validate our theoretical analysis by following the research questions:

**RQ1: Effectiveness.** Compared with generator-only sampling, does the generator–refiner pipeline produce higher-quality conformers, preserve diversity, and do so with fewer steps? (Sections 5.2 and 5.3)

**RQ2: Refiner impact.** For upstream-generated conformers, what proportion are *improved* versus *downgraded* by the refiner? (Section 5.4)

**RQ3: Sampling dynamics.** What dynamics does the refiner exhibit, and do these dynamics align with our theoretical analysis (as assessed by empirical fits)? (Section 5.5)

RQ1–RQ2 focus on performance comparisons, whereas RQ3 examines sampling dynamics and associated property behavior.

### 5.1 SETUP

**Dataset.** We evaluate on the GEOM dataset (Axelrod & Gomez-Bombarelli, 2022). GEOM-DRUGS is the largest relevant subset (304k drug-like molecules). We also train and evaluate on GEOM-QM9, a more established benchmark with smaller molecules. We follow Ganea et al. (2021) random splits of 80%/10%/10% into train/validation/test. Following Ganea et al. (2021); Jing et al. (2022), we use the same 1,000 random test molecules from the test set. The dataset splits are 106,586/13,323/1,000 (GEOM-QM9) and 243,473/30,433/1,000 (GEOM-DRUGS) molecules.

**Implementation Detail and Baseline.** We implement our refiner by fine-tuning the open-source ET-Flow (Hassan et al., 2024) architecture and weights. For fine-tuning, we reduce the learning rate from 0.007 to 0.001 and adopt a Cosine-Warmup learning rate schedule; otherwise, architectural choices and hyperparameters follow ET-Flow. At the sampling process, the refiner serves as a post-processor on conformers produced by the three most recent conformer generative models based on denoising: MCF (Wang et al., 2024), ET-Flow (Hassan et al., 2024), and DMT (Liu et al., 2025). For upstream sampling, we use the official open source code implementations and trained weights. We compare the final performance with recent advanced models (Ganea et al., 2021; Xu et al., 2022; Jing et al., 2022; Wang et al., 2024; Hassan et al., 2024; Liu et al., 2025).

**Evaluation Metric.** Following Ganea et al. (2021); Jing et al. (2022), for each molecule with $K$ reference conformers, we generate $2K$ candidate conformers. We report Average Minimum RMSD (AMR)-precision (quality), AMR–recall (diversity), and Coverage (COV) (see the Appendix A.2 for details). Following Jing et al. (2022), we also evaluate chemical similarity using properties computed with xTB (Bannwarth et al., 2019): total energy $E$, dipole moment $\mu$, HOMO–LUMO gap $\Delta\epsilon$, and minimum energy $E_{\min}$. Finally, because our refiner aims to improve conformers, we additionally report improvement and downgrade rates relative to each conformer's baseline quality.

### 5.2 ENSEMBLE RMSD AND SAMPLING EFFICIENCY

To demonstrate higher quality with fewer sampling steps, we adopt a stricter budget: baselines use a single generator with 50 sampling steps, while our pipeline (generator plus refiner) uses 40 steps in total (20 for generation and 20 for refinement). Following the original papers, we run MCF and DMT at their largest-scale configurations: Large (L) on GEOM-DRUGS and Basic (B) on GEOM-QM9.

As shown in Table 1 and Table 2, our method surpasses baselines. On the precision metric (AMR), the median on GEOM-DRUGS decreases by 10.94%, and on GEOM-QM9, the error is roughly halved. On the diversity metric (recall), we see a smaller improvement: 8.36% on GEOM-DRUGS AMR-median and 14.6% on GEOM-QM9 AMR-median. These recall gains are driven mainly by improving conformer quality: for each ground truth conformer, the generated conformer that was previously its nearest match remains the match but fits better after refinement. Thus, the refiner substantially improves quality and preserves diversity. See the Appendix A.3 for additional ablations on the number of steps.

Table 4: Improvement rate (IR) / Downgrade rates (DR) (%) using RMSD Precision with a relative tolerance $\tau$ (Å) of GEOM-Drugs

| $\tau$ | MCF $\rightarrow$ + Refiner | | ET-Flow $\rightarrow$ + Refiner | | DMT $\rightarrow$ + Refiner | |
|---|---|---|---|---|---|---|
| | IR | DR | IR | DR | IR | DR |
| 0.05 | 37.5 | 16.7 | 35.3 | 5.6 | 22.7 | 7.5 |
| 0.10 | 23.9 | 7.0 | 15.7 | 1.2 | 9.7 | 1.8 |
| 0.20 | 9.6 | 1.4 | 2.6 | 0.1 | 2.1 | 0.2 |
| 0.50 | 0.5 | 0.0 | 0.0 | 0.0 | 0.0 | 0.0 |

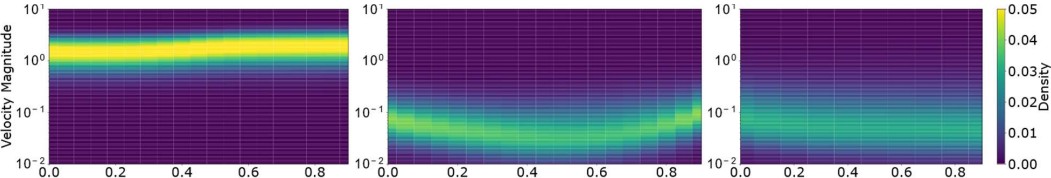

Figure 4: Velocity fields on GEOM–QM9: (a) ET–Flow sampling; (b) Refiner; (c) refiner with randomized $t$.

## 5.3 CHEMICAL PROPERTY

We also compare the chemical similarity between generated and ground truth conformers. We follow and use the same 100-molecule subset of Jing et al. (2022). For a molecule with $K$ ground truth conformers, we randomly select $\min(2K, 32)$ generated conformers, relax them with GFN2-xTB(Bannwarth et al., 2019), and compare Boltzmann-weighted ensemble properties between the generated and ground truth sets. Using xTB(Bannwarth et al., 2019), we compute energy ($E$), dipole moment ($\mu$), HOMO–LUMO gap ($\Delta\varepsilon$), and minimum energy ($E_{\min}$). Table 3 reports median errors, showing that our method can get better chemically accurate ensembles.

Table 3: Median Boltzmann-weighted errors of ensemble properties between sampled and generated conformers: $E$, $\Delta\varepsilon$ in kcal/mol, and $\mu$ in debye; median and $E_{min}$ in kcal/mol.

| | $E$ | $\mu$ | $\Delta\epsilon$ | $E_{\min}$ |
|---|---|---|---|---|
| GeoDiff | 0.31 | 0.35 | 0.89 | 0.39 |
| GeoMol | 0.42 | 0.34 | 0.59 | 0.40 |
| Torsional Diff. | 0.22 | 0.35 | 0.54 | 0.13 |
| MCF-L | 0.68 | 0.28 | 0.63 | 0.04 |
| ET-Flow | 0.23 | 0.19 | **0.38** | 0.02 |
| MCF-L + **Refiner** | **0.20** | 0.23 | **0.38** | 0.02 |
| ET-Flow + **Refiner** | 0.21 | **0.18** | 0.39 | **0.01** |

## 5.4 IMPROVEMENT AND DOWNGRADE RATE ON RMSD ENSEMBLE

Beyond the macro-level improvement in Section 5.2, we test at a micro level whether the refiner improves conformers for different upstream generative models. For every conformer, we compute precision RMSD before and after refinement and apply multi-tolerance thresholds $\tau$ (Å) to label outcomes as improvement or downgrade. Each conformer is paired only with its own refined counterpart (one-to-one). We then report improvement/downgrade rates for each group in Tables 4 and 7. Across all comparisons and thresholds, our refinement yields improvement rates that are multiples of the downgrade rates.

## 5.5 EMPIRICAL EXAMINATION OF SAMPLING DYNAMICS

We empirically examine the sampling dynamics induced by the properties introduced in Section 4.2, which drive the refiner. To avoid architectural side effects, the upstream is fixed to ET-Flow (Hassan et al., 2024), which is the refiner fine-tuning on. We use GEOM–QM9, whose stronger upstream quality implies a larger $(\mathbf{x}_t, t)$ vs. $(\mathbf{x}_{t^\star}, t^\star)$ mismatch, making it a stricter test.

**Self-calibration and two stages.** Fig. 5 traces the RMSD throughout refinement. A short *warm-up* period ($t \in [0, 0.3]$) exhibits a small RMSD increase due to neighbor-degree mismatch; then, once the neighbor degrees are aligned better under the $t$-schedule, RMSD decreases monotonically during

*refinement*. At $t = 1$, the median precision AMR declines from $\approx 0.05$ (Å) to $\approx 0.03$ (Å), with a noticeable subset attaining RMSD $< 0.01$ (Å), which is rarely observed with a single flow-matching model.

**Velocity dynamics.** The second key property is the velocity behavior. In Fig. 4, (a) ET-Flow shows typical speeds exceeding 1; (b) the refiner concentrates below $\sim 0.1$; and (c) when we randomize the input $t$ to the refiner, the low-velocity pattern persists. These observations indicate: when the upstream conformer starts in a relatively low-noise state, even under time-step mismatch, the refiner produces low velocities that avoid catastrophic errors during the warm-up stage. Also, compared with the much higher velocity during the sampling of the generation process, such low velocities by the refiner can keep atoms within their current basins and thereby preserve diversity.

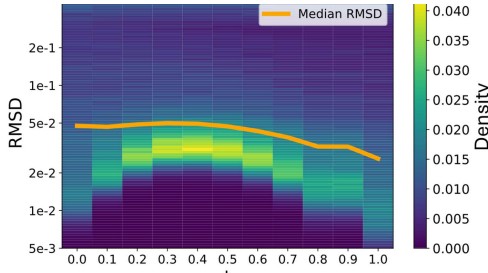

Figure 5: GEOM-QM9 AMR–precision dynamics during refinement

## 6 CONCLUSION

In this paper, we propose a flow-matching based refiner for molecular conformer generation. At sampling time, the refiner samples directly on conformers generated by diverse upstream denoising models, re-aligning the perturbed conformers on the fly without requiring access to the upstream model. By rescheduling the noise scale, the method bypasses the ill-trained low-SNR regime and early-step error propagation, yielding a clear second-stage quality gain. Empirically, with fewer total steps, our denoising model + refiner pipeline achieves better performance, and the improvement rate substantially exceeds the downgrade rate.

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

Table 5: Performance of generated conformer ensembles on the GEOM-DRUGS test set, reported as Coverage (COV, %) and Average Minimum RMSD (AMR, Å). Coverage (COV) is computed with threshold $\delta = 0.75$ Å. The best performance is **bold**.

| Method | steps | Recall | | | | Precision | | | |
|---|---|---|---|---|---|---|---|---|---|
| | | COV ↑ | | AMR ↓ | | COV ↑ | | AMR ↓ | |
| | | Mean | Med | Mean | Med | Mean | Med | Mean | Med |
| MCF-L | 10 | 84.52 | 92.86 | 0.444 | 0.408 | 65.63 | 68.75 | 0.668 | 0.608 |
| MCF-L | 20 | 85.82 | 93.18 | 0.400 | 0.369 | 66.43 | 69.54 | 0.637 | 0.560 |
| MCF-L | 50 | 85.10 | 92.86 | 0.390 | 0.343 | 66.63 | 70.00 | 0.623 | 0.546 |
| MCF-L | 100 | 85.06 | 92.12 | 0.388 | 0.348 | 66.70 | 70.45 | 0.621 | 0.539 |
| MCF-L + Refiner | 10 + 10 | 85.61 | 93.67 | 0.386 | 0.348 | 71.82 | 76.67 | 0.559 | 0.493 |
| MCF-L + Refiner | 20 + 20 | **86.44** | **93.68** | **0.368** | **0.330** | **72.07** | **78.41** | **0.550** | **0.480** |
| DMT-L | 10 | 85.82 | 91.99 | 0.391 | 0.364 | 66.62 | 70.50 | 0.625 | 0.564 |
| DMT-L | 20 | 85.58 | 92.44 | 0.384 | 0.358 | 67.48 | 71.65 | 0.608 | 0.532 |
| DMT-L | 50 | 85.95 | 91.98 | 0.378 | 0.353 | 67.97 | 71.97 | 0.599 | 0.529 |
| DMT-L | 100 | 85.80 | 92.30 | 0.375 | 0.346 | 67.90 | 72.50 | 0.598 | 0.527 |
| DMT-L + Refiner | 10 + 10 | **87.66** | 93.45 | 0.352 | 0.329 | 74.92 | 80.97 | 0.512 | 0.459 |
| DMT-L + Refiner | 20 + 20 | 87.47 | **94.12** | **0.349** | **0.319** | **75.91** | **81.51** | **0.497** | **0.446** |
| ET-Flow | 10 | 79.41 | 85.02 | 0.467 | 0.439 | 72.33 | 78.77 | 0.577 | 0.515 |
| ET-Flow | 20 | 79.28 | 84.74 | 0.467 | 0.437 | 73.34 | 80.00 | 0.562 | 0.499 |
| ET-Flow | 50 | 80.15 | 85.71 | 0.458 | 0.429 | 73.89 | 80.56 | 0.556 | 0.494 |
| ET-Flow | 100 | 79.78 | 84.15 | 0.462 | 0.436 | 73.70 | 80.00 | 0.561 | 0.504 |
| ET-Flow + Refiner | 10 + 10 | 80.20 | **85.71** | 0.445 | 0.411 | 73.47 | 81.25 | 0.547 | 0.477 |
| ET-Flow + Refiner | 20 + 20 | **80.29** | 85.11 | **0.439** | **0.410** | **74.58** | **81.59** | **0.530** | **0.467** |

# A  APPENDIX

## A.1  RMSD UPPERBOUND WITH NOISE SCALE

After removing six SE(3) degrees of freedom, the non-rigid subspace has $d := 3N - 6$ degrees of freedom. we approximate the error $\Delta$ as isotropic Gaussian noise in coordinate space with unknown scale $\sigma^\star$, we can get:

$$\|\Delta\|^2 \sim \sigma^{\star 2} \chi_d^2, \tag{13}$$

where $\chi_d^2$ is a chi-square random variable with $d$ degrees of freedom.

Consequently,

$$\mathrm{RMSD}(\widetilde{\mathbf{x}}, \mathbf{x}_1) = \sigma^\star \sqrt{\frac{1}{N}\chi_d^2} \tag{14}$$

By the Wilson–Hilferty approximation (Wilson & Hilferty, 1931), we have:

$$\left(\frac{X}{d}\right)^{1/3} \approx \mathcal{N}\left(1 - \frac{2}{9d}, \frac{2}{9d}\right) \quad \text{for } X \sim \chi_d^2. \tag{15}$$

by Eq.14, since $\mathrm{RMSD}^2 = (\sigma^{\star 2}/N)\,X$ with $X \sim \chi_d^2$,

$$\mathrm{RMSD} \approx \sigma^\star \sqrt{\frac{d}{N}\left(1 - \frac{2}{9d} + z\sqrt{\frac{2}{9d}}\right)^3}, \qquad z \sim \mathcal{N}(0,1). \tag{16}$$

Hence, denoting the standard-normal quantile at confidence level $k$ by $Q_k$, the RMSD quantile at scale $\sigma^\star$ is:

$$\mathrm{RMSD}_{(Q_k,\sigma^\star)} \approx \sigma^\star \sqrt{\frac{d}{N}\left(1 - \frac{2}{9d} + Q_k\sqrt{\frac{2}{9d}}\right)^3}. \tag{17}$$

Table 6: Performance of generated conformer ensembles on the GEOM-QM9 test set, reported as Coverage (COV, %) and Average Minimum RMSD (AMR, Å). Coverage (COV) is computed with threshold $\delta = 0.05$ Å. The best performance is **bold**.

| Method | steps | Recall | | | | Precision | | | |
| | | COV ↑ | | AMR ↓ | | COV ↑ | | AMR ↓ | |
| | | Mean | Med | Mean | Med | Mean | Med | Mean | Med |
|---|---|---|---|---|---|---|---|---|---|
| MCF-B | 10 | 51.60 | 50.00 | 0.115 | 0.067 | 43.57 | 34.00 | 0.144 | 0.094 |
| MCF-B | 20 | 62.13 | 62.50 | 0.108 | 0.056 | 55.73 | 50.00 | 0.124 | 0.068 |
| MCF-B | 50 | 66.83 | 67.86 | 0.101 | 0.050 | 61.18 | 64.29 | 0.117 | 0.059 |
| MCF-B | 100 | 68.90 | 75.00 | 0.099 | 0.047 | 64.00 | 69.12 | 0.112 | 0.052 |
| MCF-B + Refine | 10+10 | 74.32 | 80.00 | **0.097** | 0.037 | 76.97 | 94.23 | **0.100** | 0.025 |
| MCF-B + Refine | 20+20 | **74.87** | **81.82** | 0.101 | **0.035** | **77.41** | **94.44** | 0.101 | **0.023** |
| DMT-B | 10 | 69.64 | 77.78 | 0.089 | 0.041 | 61.37 | 66.29 | 0.118 | 0.062 |
| DMT-B | 20 | 71.71 | 80.00 | 0.088 | 0.038 | 66.21 | 75.00 | 0.109 | 0.048 |
| DMT-B | 50 | 72.90 | 83.33 | 0.087 | 0.036 | 67.76 | 75.00 | 0.107 | 0.047 |
| DMT-B | 100 | 73.75 | 83.33 | 0.085 | 0.036 | 68.39 | 75.00 | 0.103 | 0.047 |
| DMT-B + Refine | 10+10 | **79.74** | 88.89 | **0.069** | 0.028 | 79.32 | 93.75 | 0.079 | 0.025 |
| DMT-B + Refine | 20+20 | 79.50 | **89.44** | 0.070 | **0.026** | **80.37** | **97.92** | **0.076** | **0.021** |
| ET-Flow | 10 | 75.07 | 87.50 | 0.082 | 0.034 | 67.49 | 73.07 | 0.126 | 0.066 |
| ET-Flow | 20 | 75.65 | 85.71 | 0.083 | 0.033 | 69.40 | 75.00 | 0.121 | 0.056 |
| ET-Flow | 50 | 75.72 | 87.23 | 0.083 | 0.031 | 70.32 | 75.00 | 0.114 | 0.053 |
| ET-Flow | 100 | 76.55 | 87.50 | 0.077 | 0.030 | 70.67 | 79.75 | 0.115 | 0.047 |
| ET-flow + Refine | 10+10 | **78.66** | **89.74** | **0.075** | **0.027** | 76.73 | 87.50 | 0.106 | 0.037 |
| ET-flow + Refine | 20+20 | 78.40 | 88.89 | 0.076 | 0.028 | **77.36** | **89.94** | **0.103** | **0.031** |

Table 7: Improvement rate (IR) and Downgrade rate (DR) (%) computed by RMSD-precision with a relative tolerance $\tau$ (Å) on GEOM-QM9. Because the upstream error scale on QM9 is smaller than on GEOM-DRUGS, we adopt a tighter tolerance.

| $\tau$ | MCF $\rightarrow$ + Refiner | | ET-Flow $\rightarrow$ + Refiner | | DMT $\rightarrow$ + Refiner | |
| | IR | DR | IR | DR | IR | DR |
|---|---|---|---|---|---|---|
| 0.02 | 69.2 | 15.3 | 29.2 | 5.3 | 50.7 | 5.7 |
| 0.05 | 45.1 | 12.5 | 15.1 | 1.7 | 28.7 | 2.2 |
| 0.10 | 29.0 | 9.8 | 7.7 | 0.4 | 14.5 | 0.6 |
| 0.20 | 15.6 | 8.0 | 3.1 | 0.1 | 5.5 | 0.1 |

## A.2 GEOMETRY METRIC

Following Ganea et al. (2021); Jing et al. (2022); Hassan et al. (2024), the following works have used the so-called Average Minimum RMSD (AMR) and Coverage (COV) for Precision(P): Quality and Recall(R): the diversity, measured when generating twice as many conformers as provided by CREST. For $K = 2L$ let $\{C_l^*\}_{l \in [1,L]}$ for groundtruth and $\{C_k\}_{k \in [1,K]}$ for generated conformer.

$$\text{COV-R} := \frac{1}{L} \left| \{l \in [1..L] : \exists k \in [1..K], RMSD(C_k, C_l^*) < \delta \} \right|$$

$$\text{AMR-R} := \frac{1}{L} \sum_{l \in [1..L]} \min_{k \in [1..K]} RMSD(C_k, C_l^*)$$

(18)

The $\delta$ is the coverage threshold, and the precision metrics are obtained by swapping ground truth and generated conformers.

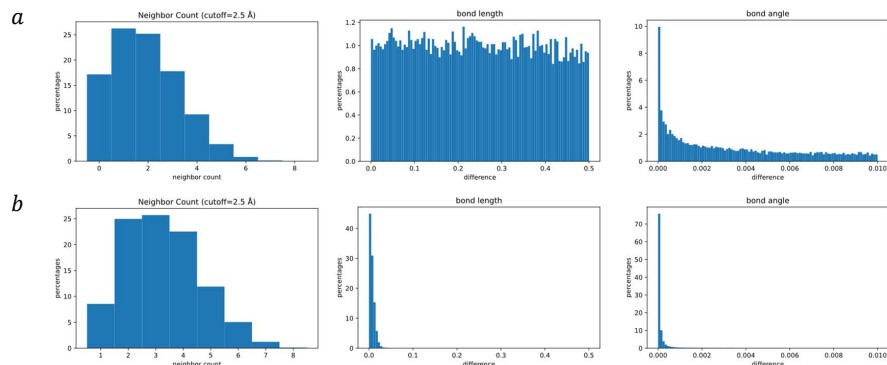

Figure 6: Comparison between (a) the training distribution of $x_0$ and (b) the test-time Refiner inputs $\hat{x}$. We visualize three geometric components: bond lengths, bond angles (measured as $1 - \cos(\Delta\theta)$), and neighbor degrees. Across all three dimensions, the test-time inputs lie well within the coverage of the training noise distribution, consistent with the analysis in Sec. 4.2.

### A.3 ADDITIONAL RESULTS: REFINER VS. MORE SAMPLING STEPS

We present a detailed ablation comparing (i) increasing the upstream sampler's steps and (ii) applying the refiner at equal or lower steps. Concretely, we evaluate base models at 10/20/50/100 steps and contrast them with "$k$ steps + refiner" ($10+10$, $20+20$). Results of Geom-Drugs can be found in Table 5 and GEOM-QM9 can be found in Table 6.

### A.4 CONFORMER-LEVEL IMPROVEMENT (QM9)

Complementing Section 5.4, we report per-conformer improvement/downgrade on GEOM–QM9. For each conformer, we compute precision RMSD before/after refinement, apply multi-tolerance thresholds $\tau$ (Å) to label **improvement** vs. **downgrade**. Details can be found in Table 7.

### A.5 TRAINING–TEST DISTRIBUTION COMPARISON

To more precisely examine the potential mismatch between the training distribution of $x_0$ and the test-time Refiner inputs $\hat{x}$, we decompose molecular geometry into three minimal primitives: (i) bond lengths (distances), (ii) bond angles (directions), and (iii) neighbor degrees. These quantities correspond directly to the distance–direction elements processed by our equivariant aggregation, and therefore provide a natural basis for comparing geometric coverage.

Figure 6 reports the empirical distributions of these three primitives under both the training noise schedule and the test-time upstream-generated inputs. Bond lengths and bond angles under test-time noise fall almost entirely within the support of their corresponding training distributions, indicating that the Refiner consistently operates on geometric patterns it has already encountered during training. Neighbor degree exhibits the only noticeable $t$-dependent variation, but the shift is small. Moreover, its expected value increases monotonically as the training noise level decreases, leading to a natural $t$-realignment effect in which the local neighborhoods gradually restore their ground-truth structure along the refinement trajectory. This variation influences only the aggregation scale and does not compromise geometric consistency.

Overall, these results confirm that the Refiner's test-time inputs remain well covered by the training noise distribution, reinforcing the stability argument presented in Sec . 4.2.

### A.6 CASE STUDY

We provide case studies to better illustrate how the Refiner behaves in practice. We first randomly sample molecules, and for each selected molecule, we identify the generated conformers that exhibit

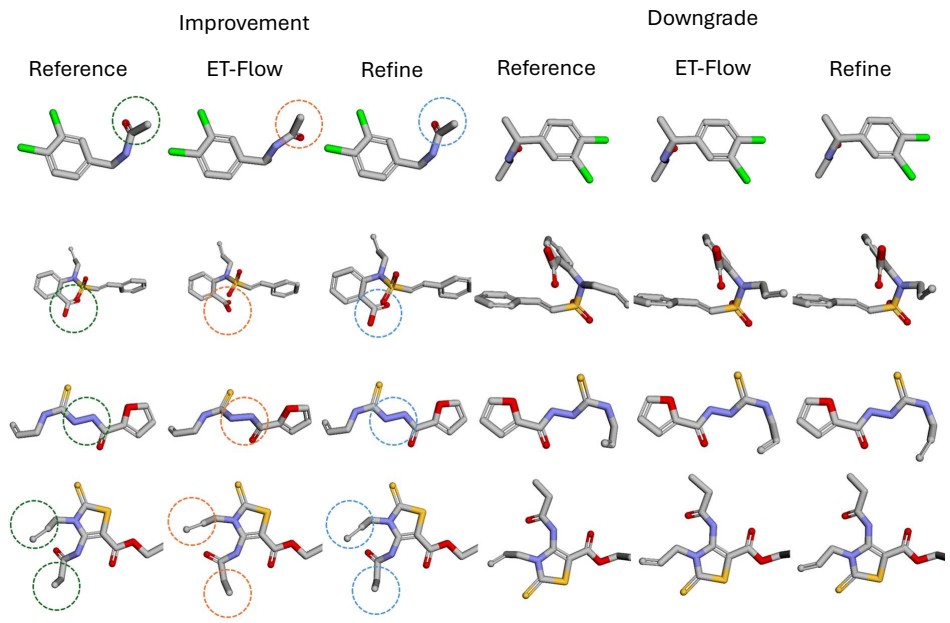

Figure 7: Qualitative visualization of the Refiner. The left column shows the conformers with the largest improvements, while the right column shows those with the largest downgrades. All conformers are generated for the same molecule. In the right column, the circles highlight the main structural differences among the reference, generated, and refined conformers.

the largest improvement and the largest degradation in RMSD after refinement. For each such conformer, we visualize the reference, the upstream generated, and the refined structures. Within the same molecule, different generated conformers therefore serve as paired examples of substantial improvements and mild downgrades.

In the improvement cases (left column of Figure 7), we observe that the upstream ET-Flow samples retain certain structural errors, such as distorted local geometries, which are subsequently corrected by the Refiner. In contrast, for the downgrade cases (right column), the refined conformers remain broadly consistent with the corresponding generated ones: there is no evidence of new systematic errors being introduced. The observed degradations tend to be small and localized, and can be attributed to minor fluctuations in model capacity rather than to any inherent instability of the refinement process.

Taken together, these observations suggest that the Refiner acts as a local geometric corrector: for reasonably good upstream conformers, it can reliably improve the structure at the local level, which in turn leads to consistent macroscopic gains in overall quality. This trend is quantified in Tables **??** and **??**, which report clear net improvements despite the presence of a small number of downgraded conformers. The visualizations further reveal that the Refiner is able to repair structural issues (e.g., clearly mismatched torsional angles), whereas the downgraded cases remain local in nature and do not introduce new structural errors.

## A.7 DIVERSITY PRESERVATION STUDY

Figure 8 further supports this observation by examining recall improvements across different RMSD bins. The gains in recall are mainly concentrated in the low-RMSD regions, highlighting the Refiner's strength as a local geometric optimizer for already reasonable conformers. For ground-truth conformers with large RMSD-based recall gaps (i.e., those far from any generated conformer), the improvement is modest, and the curves remain nearly unchanged before and after refinement.

Crucially, we do not observe any systematic degradation in recall. The Refiner does not collapse multiple generated conformers onto a small number of ground-truth conformers, nor does it in-

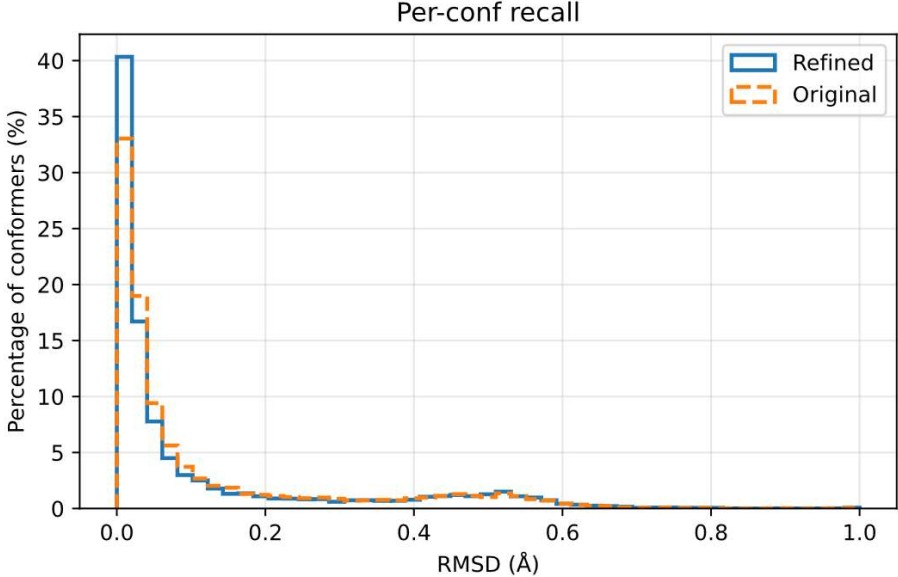

Figure 8: Per-conformer recall distribution before and after refinement, grouped by RMSD bins.

duce oversmoothing of the conformational ensemble. This indicates that the Refiner consistently improves conformer quality while preserving the overall diversity of the generated set.

