# OpenReview forum: "Flow-Matching based Refiner for Molecular Conformer Generation"
_ICLR.cc/2026/Conference — Submitted to ICLR 2026_

### Official Review · Reviewer_Xj5K · 2025-10-25

**Soundness:** 3
**Presentation:** 3
**Contribution:** 2
**Rating:** 4
**Confidence:** 4

**Summary:**

This paper proposes a flow-matching-based refiner, coupled with existing denoising models, to improve molecular conformer generation.

In particular, the proposed refiner directly starts from conformers generated by diverse upstream denoising models,  thereby skipping the high-noise phase and improving final performance.

Experiments demonstrate that the proposed pipeline (generator plus refiner) achieves higher quality with fewer total sampling steps.

**Strengths:**

1. The proposed refiner is plug-and-play, compatible with conformers generated by diverse upstream denoising models, and requires no model-specific tuning.

2. Compared with baselines using a single denoising model, employing the proposed refiner as the post-processor improves performance while reducing sampling steps.

3. The whole paper is excellently written.

**Weaknesses:**

1. As shown in Algorithm 1, while the refiner is trained using the flow-matching paradigm, $x_0$ is directly derived from $x_1$. This establishes a deterministic mapping between them, rather than a genuine distributional transport. In this case, the refiner functions more as a conditional corrective regressor than as a genuine flow field that transports one distribution to another.

2. Essentially, the proposed refiner represents a form of learned refinement, which makes it necessary to compare with other refinement methods, especially those typical optimizers like MMFF and UFF. The absence of such comparisons leaves unanswered questions regarding the refiner's effectiveness relative to established techniques.

3. The paper lacks an officially recommended reproducibility statement, which is crucial for ensuring the credibility of this work. Moreover, the absence of code availability further hinders the ability to reproduce the presented results.

**Questions:**

1. Could you clarify why Algorithm 1 does not align with Equation 6?

2. Could you clarify the rationale behind directly implementing the proposed refiner by fine-tuning on ET-Flow? What would be the implications of training from scratch? Additionally, how might the results vary if different architectures were utilized?

3. Could you describe the use of Large Language Models (LLMs) in your paper? According to the ICLR 2026 Author Guide, failure to disclose significant usage of Large Language Models (LLMs) may result in the desk rejection of the paper.

---

> ### Author Response · Authors · 2025-11-21
>
> Thank you for your valuable feedback. Below, we provide our responses to the questions and weaknesses you raised. We hope our clarifications adequately address your concerns.
>
> **W1/Q1 Inconsistency between Algorithm 1 vs. Eq. (6)–(9):** Thank you for pointing out this inconsistency. you are correct that the line $u_t = x_1 - x_0$ in Algorithm 1 does not match Eq. (6)–(9). This line is a small typo in the pseudocode and corresponds to the special case $s(t) = 0$; it does not reflect our actual implementation, which follows the full $t$-dependent objective in Eq. (6)–(9). To clarify the influence of $s(t)$, we report an ablation on QM9 comparing the default schedule $s(t)$ with the simplified choice $s(t) = 0$:
>
> | QM9      | Recall-COV↑ || Recall-AMR↓ || Precision-COV↑ || Precision-AMR↓ ||
> |----------|------------|--------|------------|--------|--------------|--------|--------|--------|
> | $s(t)$ | Mean       | Median | Mean       | Median | Mean         | Median |Mean         | Median |
> | default  | **78.66%** | **89.74%** | **0.075** | **0.027** | **76.73%** | **87.50%** | **0.105** | **0.037** |
> | 0        | 78.46%     | 88.89% | 0.076      | **0.027** | 76.26%       | 85.50% | 0.107      | 0.038     |
>
> As shown, setting $s(t) = 0$ shows only minor differences compared to the default $s(t)$ schedule. This indicates that the core behavior and conclusions of the Refiner are not sensitive to the specific choice of $s(t)$. We have corrected Algorithm 1 in the revised version to ensure full consistency with Equations (6)–(9).
>
> **W2 Comparison to Classical Refinement Methods:**  Classical refinement methods, such as MMFF and UFF are based on hand-crafted molecular mechanics force fields that approximate the potential energy surface with fixed functional forms and parameter sets, rather than learning directly from the data distribution. As a result, they are often no longer sufficient to further optimize the conformers produced by state-of-the-art generative models: for example, as shown in the following table, applying MMFF post-optimization to ET-Flow outputs on GEOM-QM9 actually worsens the RMSD-based metrics, whereas our learned Refiner consistently improves them.
>
> | QM9              |Recall-COV↑||Recall-AMR↓||Precision-COV ↑||Precision-AMR↓||
> |-|-|-|-|-|-|-|-|-
> | s(t)             |Mean |Median|Mean |Median|Mean|Median| Mean |Median|
> | ET-Flow  |75.07%	|87.50%	|0.082	|0.034 | 67.49%|	73.07%|	0.126	|0.066|
> |  +MMFF | 16.83%|0%|0.182|0.135|15.23%|0%|0.187|0.147|
> |  +Refiner | **78.46%**|**88.89%**|**0.076**|**0.027**|**76.26%**|**85.50%**|**0.107**|**0.038**|
>
> **W3 Reproducibility:** We provide anonymous code in the supplementary materials
>
> **Q2 Design Choices: Fine-Tuning Strategy and Architectural Generality:** We chose to implement the Refiner by fine-tuning an ET–Flow–style architecture primarily because ET-Flow is a strong, lightweight SE(3)-equivariant baseline that is already well-validated on GEOM-QM9/GEOM-DRUGS, and our focus in this work is on the refinement mechanism itself rather than proposing yet another new backbone. In principle, training the Refiner from scratch with the same architecture would not alter the core methodology; we expect similar behavior, but at a higher computational cost, since the model would need to relearn the geometric features that ET-Flow already captures. Likewise, the Refiner is not conceptually tied to ET-Flow: any reasonable SE(3)-equivariant architecture (including those used in DMT or MCF) could be used as the backbone. Systematically exploring alternative architectures is not related to our main contribution and is left as future work.
>
> **Q3 LLM Usage Disclosure:** LLMs were used only for language polishing and grammar refinement during the writing process. All methodological formulations, theoretical motivations, and evaluations were developed entirely by the authors.

---

### Official Review · Reviewer_jEgW · 2025-10-30

**Soundness:** 3
**Presentation:** 2
**Contribution:** 2
**Rating:** 4
**Confidence:** 4

**Summary:**

The paper addresses the task of molecular conformer generation, where models often suffer from error accumulation in early high-noise stages. To mitigate this, the authors introduce a flow-matching–based refiner that starts from existing conformers instead of pure noise, effectively bypassing low-SNR phases. This approach enhances both conformer quality and sampling efficiency, producing more accurate and diverse molecular structures with fewer steps.

**Strengths:**

- Introduces a flow-matching–based refiner that improves molecular conformer generation (MCG) by refining upstream outputs instead of starting from pure noise.
- Bypasses the low signal-to-noise ratio (SNR) regime and reduces early-step error accumulation typical in denoising-based models.
- Improves precision without collapsing conformer diversity.
- Provides theoretical justification for noise scale rescheduling, robustness to time-step mismatch, and diversity preservation.

**Weaknesses:**

- Code is not available.
- Although the refiner shortens sampling time, the paper does not clarify its memory overhead or overall computational trade-offs relative to the achieved performance gains.
- The refiner assumes that upstream conformers are reasonably accurate structures with small random deviations. However, its performance when the upstream samples are systematically biased or incorrect remains unclear.
- The paper lacks qualitative analysis or visualization of instances where the refiner degrades conformer quality, which limits understanding of its robustness.
- The paper does not include a clear schematic or main figure illustrating the overall model architecture, which makes it difficult for readers to understand how the refiner integrates with the upstream generator and the sampling process.

**Questions:**

- How robust is the refiner when upstream conformers contain systematic structural biases or chemically incorrect geometries rather than small random perturbations?
- In what specific cases does the refiner fail or degrade conformer quality, and what underlying factors contribute to these failure modes?
- How does the proposed refiner balance reduced sampling time against potential increases in memory consumption or computational cost, and what are the quantitative trade-offs involved?

---

> ### Author Response · Authors · 2025-11-21
>
> Thank you for your valuable feedback. Below, we provide our responses to the questions and weaknesses you raised. We hope our clarifications adequately address your concerns.
>
> **W1 Reproducibility:** We provide anonymous code in the supplement materials
>
> **W2 / Q3 Computational Cost and Memory Usage:** At sampling time, our end-to-end pipeline is purely sequential: Gaussian noise → diffusion/flow-matching generator → upstream conformer → Refiner → final high-quality conformer. All stages are executed one after another, and we do not retain intermediate activations from multiple models in memory; therefore, the peak memory usage remains essentially unchanged.
> Theoretically, for ET-Flow + Refiner, a single Refiner forward pass has a similar per-step cost to one ET-Flow step; however, it allows us to shorten the upstream trajectory to 10 steps, resulting in a net reduction in wall-clock sampling time. More detailed ablations over the number of sampling steps are provided in Tables 6 and 7: the 10 + 10 configuration (10 upstream steps + 10 Refiner steps) already surpasses the performance of using 100 upstream-only steps.
> Following previous work, we quantify computational cost using wall-clock sampling time under a fixed hardware and batch configuration. All methods are evaluated on an NVIDIA A100 with batch size = 64, using 10 sampling steps and their recommended numerical precision:
>
> | Method             | Average time (seconds) | Precision |
> |--------------------|------------------------|-----------|
> | ET-Flow / Refiner  | 0.03                   | FP32      |
> | MCF-L              | 0.07                   | FP32      |
> | DMT-L              | 0.04                   | FP16      |
>
> Under the same number of steps, ET-Flow and the Refiner achieve the lowest wall-clock cost among these baselines. This is expected because both methods share the same architectural structure, including the same PC sampling strategy; the only differences lie in their model weights and the initial input coordinates (which have the same shape but different values). Therefore, using the number of sampling steps as a proxy for computational cost is a conservative and fair way to evaluate the refiner.
>
> **W3/Q1 Robustness:** To simulate more systematic and larger structural errors, we also use RDKit-generated conformers as upstream inputs. RDKit conformers are well known to exhibit geometric drift from real structures, thus providing a stronger stress test for the Refiner.
> As shown below, the raw RDKit conformers yield extremely poor recall/precision, confirming that this setting indeed corresponds to a much harder error distribution. Despite this, applying the Refiner dramatically improves all metrics:
> | QM9            |Recall-COV||Recall-AMR||Precision-COV ||Precision-AMR||
> |-|-|-|-|-|-|-|-|-
> | method    |Mean |Median|Mean |Median|Mean|Median| Mean |Median|
> | rdkit		| 2.73%|0.00%	|0.2313|0.1728|1.73%|0.00%		|0.5120|0.4878|
> |  rdkit + refiner	|77.88%|87.50%	|0.0780|0.0294|76.02%|83.33%	|0.1071|0.0395|
>
> Notably, after refinement, the performance becomes comparable to or even better than the diffusion/flow-matching model. This demonstrates that the Refiner is highly robust even under large, systematic upstream errors, and that its improvements are not limited to small perturbations
>
> **W4/Q2 Case Studies, and Visual Analysis:**
> In the appendix 6 (Fig 7), we have added new case studies that explicitly visualize molecules where the Refiner both improves and degrades different generated conformers. We select extreme examples in which, for the same molecule, one conformer exhibits a large downgrade while another shows a larger improvement. However, even in these examples, the magnitude and frequency of degradations are lower than those of the improvements, suggesting that such downgrades are rare and the Refiner remains robust overall.
>
> **W5 Pipeline Illustration:**
> We add an overview figure (fig2) of the proposed refiner pipeline

---

### Official Review · Reviewer_CRoH · 2025-10-30

**Soundness:** 3
**Presentation:** 3
**Contribution:** 3
**Rating:** 6
**Confidence:** 4

**Summary:**

This paper proposes a "flow-matching based refiner" architecture for molecular conformer generation (MCG) tasks. The approach addresses a key limitation in denoising generative models—error accumulation from early, high-noise steps—by introducing a refiner model that operates on outputs from upstream generative methods, skipping the problematic low-SNR regime through noise rescheduling and leveraging flow-matching with a carefully chosen base distribution. The refiner is shown to be plug-and-play with different upstream methods and is validated on the GEOM-DRUGS and GEOM-QM9 benchmarks, exhibiting improvements in sample quality (as measured by AMR and coverage), sample diversity, and chemical property alignment, even with fewer sampling steps.

**Strengths:**

1. The method’s theoretical justification is detailed, with a concrete analysis of the implications of time-step mismatch and noise scale alignment. The derivation (see Equations in Section 4.1 and Algorithm 1) is mathematically sound and elucidates how the refiner provides a principled (rather than heuristic) way to bypass the ill-behaved regime in diffusion/flow models.
2. Experimental results are compelling: Tables 1 and 2 show consistent improvement in AMR and coverage on both GEOM-DRUGS and GEOM-QM9, even when using fewer total sampling steps than the baseline models. Table 3 further demonstrates enhanced performance in chemical property prediction.
3. The paper offers ablation studies and broad benchmarking, including effects of sampling steps (Table 5, Table 6) and per-conformer improvement/downgrade analysis (Table 4, Table 7), supporting the claim that the refiner meaningfully improves MCG robustness and output quality.

**Weaknesses:**

1. The mathematical justification for the noise scale self-calibration is partially heuristic. While the paper introduces $t^* = 1 - \sigma^*/\sigma$ and suggests the schedule handles value-range coverage, this only provides an approximate match between upstream and refiner noise regimes. There is no guarantee that in complex distributions this calibration will avoid suboptimal updates, nor is the implications of possible misalignment deeply discussed; this is a potential source of instability or model non-robustness, especially in molecules with unusual geometry or heavy tails in error distributions.
2. Scope of empirical studies is still centered on established (or highly similar) datasets (GEOM-DRUGS, GEOM-QM9). While the results are strong, the generalizability claim ("plug-and-play", "needs no per-model tuning") is not fully supported for more diverse, realistic or larger-scale datasets (e.g., proteins, or non-Geom drug-like molecules). There are no results or analyses regarding transferability to domains with different data distributions or conformational complexity.
3. Absence of direct ablation between refiner, re-tuned upstream models, and truly hybrid or joint training strategies. In Table 1 and Figure 1, the comparison is between using more sampling steps in the generator alone versus adding a refiner with less total steps. While this demonstrates efficiency, it leaves open the question whether re-training the upstream generator to exploit more sampling steps (while increasing capacity or tuning the noise schedule) could close the gap, or if the refiner's effect would be additive if trained jointly. These are closely related to the claims of plug-and-play and step-efficiency.

**Questions:**

See the weaknesses.

---

> ### Author Response · Authors · 2025-11-21
>
> Thank you for your valuable feedback. Below, we provide our responses to the questions and weaknesses you raised. We hope our clarifications adequately address your concerns.
>
> **W1.  Self-calibration Design and Robustness:**
> **a). Calibration** The noise calibration in our method is not arbitrary. The Refiner’s update rule (Sec 4.2) depends only on three local geometric factors: pairwise distances, relative directions, and the neighbor degree. So, these are the theoretically relevant dimensions along which the training distribution must cover the upstream residuals. Our noise range only needs to ensure broad coverage along these dimensions, rather than to match any particular residual model. To validate this, Appendix A5 compares the distributions of bond lengths (distance), bond angles (direction), and neighbor degrees under training noise and under upstream-generated error. Distances and directions are fully covered by the training distribution; the only shift occurs in the neighbor degree, which is mild and monotonic in $t$, and can be realigned during refinement (Fig.3). Thus, even though the calibration is not provably optimal, it is not heuristic: it follows from the model’s geometric structure, and the empirical measurements confirm that the upstream residuals lie within the training-time coverage.
> **b). Robustness**: To simulate more systematic and larger structural errors, we also use RDKit-generated conformers as upstream inputs. RDKit conformers are well known to exhibit significant geometric drift from reference structures, with highly anisotropic and non-Gaussian residuals, and thus provide a much stronger stress test for the Refiner. As shown below on GEOM-QM9, raw RDKit conformers yield extremely poor recall and precision, confirming that this setting indeed corresponds to a substantially harder error distribution; nevertheless, applying the Refiner dramatically improves all metrics:
> |QM9   |Recall-COV↑||Recall-AMR↓||Precision-COV ↑||Precision-AMR↓||
> |-|-|-|-|-|-|-|-|-
> | method    |Mean |Median|Mean |Median|Mean|Median| Mean |Median|
> | rdkit		| 2.73%|0.00%	|0.2313|0.1728|1.73%|0.00%		|0.5120|0.4878|
> |  rdkit + refiner	|77.88%|87.50%	|0.0780|0.0294|76.02%|83.33%	|0.1071|0.0395|
>
> **W2.On Scope and Generalizability:**
> Our work is explicitly scoped to the molecular conformer generation (MCG) task, and GEOM-QM9 / GEOM-DRUGS are the standard benchmarks used by all upstream generators we build upon (ET-Flow, DMT-L, MCF-L). Our “plug-and-play’’ claim is therefore made within this task: a single Refiner architecture can be attached to heterogeneous pre-trained MCG generators without per-model tuning and consistently improves their outputs on these canonical datasets. Extending Refiner methods to proteins or non-GEOM drug-like molecules is an interesting direction, but requires new models and datasets, and is beyond the scope of this paper.
>
> **W3. Step Efficiency, and Fairness of the Comparison:**
> Appendix Tables 6–7 provide step-matched ablations: e.g., 10 upstream + 10 refiner steps already outperform 100 upstream-only steps, supporting our claim of step efficiency. Our experiments follow the standard diffusion/flow-matching setting used in prior diffusion-based MCG work: the upstream generator and the refiner are both trained with a continuous time variable $t\in(0,1)$, so different test-time step counts are simply different discretizations of the same trajectory; within this framework, re-training or jointly training new upstream models for each step budget would correspond to changing the overall system rather than ablation study of the refiner itself.

---

### Official Review · Reviewer_WFCh · 2025-10-31

**Soundness:** 3
**Presentation:** 3
**Contribution:** 3
**Rating:** 4
**Confidence:** 3

**Summary:**

This paper proposes a method in molecular conformation generation (MCG) that cascades an upstream denoising generation model (diffusion or flow-matching) with a "Refiner". Instead of starting from pure noise, the Refiner uses the conformation of the upstream model as its initial state and bypasses the low signal-to-noise ratio (low-SNR) phase by resetting/aligning the noise scale, thereby reducing early error accumulation and aiming to achieve better geometric accuracy and chemical property consistency with fewer total steps. The paper compares the benefits of allocating budget to "adding steps" versus "integrating the Refiner" in GEOM-Drugs and GEOM-QM9, and reports improvements in AMR/COV and several property metrics.

**Strengths:**

1.To address the challenges of learning low SNR steps, error propagation, and amplification in the early stages of denoising, we propose an intuitive and engineering implementation method that "skips the pure noise stage," aligning with recent error propagation analysis.

2.Refiner can theoretically be used "plug and play" to perform post-processing on various upstream model outputs, making it highly valuable for practical applications.

3.Comparing multiple strong baselines (MCF/ET-Flow/DMT) with two major benchmarks (QM9/Drugs), the report presents ensemble indices and chemical property results, and provides per-conformer improvement and degradation ratios.

4.Comparing the benefits of "increasing the number of upstream sampling steps" with "integrating Refiner", the study emphasizes the scenario where a better median AMR and comparable COV are achieved with fewer total steps.

**Weaknesses:**

1.The main text introduces an explicit t-dependency between the random interpolation term s(t)z and u_t = -sigma varepsilon + s'(t)z in the Refiner's interpolation and target velocity; however, the training objective of Algorithm 1 is written as u_t \leftarrow x_1 - x_0, which is completely independent of t and s(t), and does not reflect the noise derivative term (it even does not conform to the notation of Eq.(7)/(8)). If Algorithm 1 is implemented, the model will regress a constant velocity, which does not conform to the Flow-Matching training paradigm of "random interpolation-conditional velocity regression", and cannot reflect the instantaneous velocity scheduling when "bypassing low SNR".

2.In real systems, upstream geometric errors often exhibit conformation-dependent/structure-dependent anisotropic distributions (coupling and constraint of bond lengths/bond angles/dihedral angles, ring strain, steric barriers, etc.). Using an isotropic Gaussian approximation may underestimate the local errors and non-Gaussian tails in difficult cases. It is recommended to provide empirical tests (e.g., covariance spectra/thick tail tests of the atom-by-atom residuals of ET-Flow/DMT outputs and real conformations) to support whether this assumption is sufficient for "t-realignment".

3.During training, a "data core" basis distribution is constructed using x_{0}=x_{1}+σε; during testing, the upstream-generated 'hat{x}' is used as x_{0}. The noise and bias distributions of these two distributions are not equivalent.

4.When comparing "50-step single model" vs. "20 + 20 (generation + refinement) = 40 steps", it is necessary to clarify whether the cost per step is comparable (whether the two-stage model structure/forward FLOPs are equivalent, and whether both include PC correction steps). If Refiner's per-step cost is lighter or heavier, an equivalent cost comparison in milliseconds/molecule or FLOPs/molecule should be provided to avoid confusion between "fewer steps" and "less computational power/latency".

5.Several increases (e.g., GEOM-Drugs COV mean +1.95%) were small, and variance/confidence intervals/multiple random seed replication experiments were missing. It is recommended to report the mean ± standard deviation or 95% CI for each curve/table, along with the number of repetitions and significance tests.

6.Fig.1 suggests adding error bars, multiple random seeds and coordinate axis units (AMR units Å), and model parameter quantity/single-step cost annotations to ensure a fair comparison of "adding steps vs. adding Refiner".

**Questions:**

1.Inconsistency between Algorithm 1 and Eq. (6–9): Was z actually sampled during training and regression performed using u_{t}=−σε+s(t)z? Or was a simplified constant objective x_{1}−x_{0} used? Please provide the final executable formula and pseudocode used, and explain the performance difference between the two.

2.In practice, is explicit estimation or search used (e.g., using early steps to locate based on velocity norm or neighborhood degree stability), or is a match simply "bumped" into a fixed schedule? If the latter, how significant is the impact of mismatch on performance, and are there any adaptive strategies? Can statistical evidence (such as per-atom residual ellipsoidal spectra, heavy-tailed tests, or molecular subsets of ring systems) be provided to support or define the applicability of this hypothesis?

3.Besides AMR-Recall/COV, were distribution metrics for conformational family coverage (such as KL/FID for conformational energy histogram/dihedral distribution, and spread for Cartesian vs. interior coordinates) evaluated? Are there any negative examples of "oversmoothing" or "mode collapse" visualized?

4.How does the total latency/total FLOPs of the two-stage system compare to that of the single-stage system under the same quality conditions? Are the number of PC corrections consistent?

5.Refiner, derived from the ET-Flow architecture/weight fine-tuning, does its performance when transferred to MCF/DMT significantly depend on the upstream error morphology? Should we try a cross-robustness table that "trains with A and refines with B/C"?

6.If we change x_{0} during training to “upstream samples + slight random perturbation”, can we further approximate the test distribution and bring robustness gains?

---

> ### Author Response · Authors · 2025-11-21
>
> Thank you for your valuable feedback. Below, we provide our responses to the questions and weaknesses you raised. We hope our clarifications adequately address your concerns.
>
> **W1/Q1  Inconsistency between Algorithm 1 vs. Eq. (6)–(9):**
> Thank you for pointing out this inconsistency. you are correct that the line $u_t = x_1 - x_0$ in Algorithm 1 does not match Eq. (6)–(9). This line is a small typo in the pseudocode and corresponds to the special case $s(t) = 0$; it does not reflect our actual implementation, which follows the full $t$-dependent objective in Eq. (6)–(9). To clarify the influence of $s(t)$, we report an ablation on QM9 comparing the default schedule $s(t)$ with the simplified choice $s(t) = 0$:
>
> | QM9      | Recall-COV↑ || Recall-AMR↓ || Precision-COV↑ || Precision-AMR↓ ||
> |----------|------------|--------|------------|--------|--------------|--------|--------|--------|
> | $s(t)$ | Mean       | Median | Mean       | Median | Mean         | Median |Mean         | Median |
> | default  | **78.66%** | **89.74%** | **0.075** | **0.027** | **76.73%** | **87.50%** | **0.105** | **0.037** |
> | 0        | 78.46%     | 88.89% | 0.076      | **0.027** | 76.26%       | 85.50% | 0.107      | 0.038     |
>
> As shown, setting $s(t) = 0$ shows only minor differences compared to the default $s(t)$ schedule. This indicates that the core behavior and conclusions of the Refiner are not sensitive to the specific choice of $s(t)$. We have corrected Algorithm 1 in the revised version to ensure full consistency with Equations (6)–(9).
>
> **W2 / W3 / Q2 Training–sampling distribution difference and t-realignment:**
> The Refiner is a coordinate-update model whose per-atom correction depends only on three local geometric elements (Sec. 4.2): pairwise distances to neighbors, relative directions, and the neighbor degree. To examine train–test mismatch in a way that is aligned with this mechanism, we compare the distributions of bond lengths, bond angles, and neighbor degrees under training noise and under upstream-generated noise (Appendix A5). We find that distances and directions under test-time noise are well covered by the training distributions, and the only visible shift is a slightly smaller neighbor degree at test time; moreover, as $t$ increases, the expected neighbor degree increases monotonically (Fig. 3), so the refinement trajectory naturally realigns it into the high-density training range. Thus, while the noise-scale calibration is not a closed-form optimum, it is not purely heuristic: it is grounded in the Refiner’s geometric update structure, and empirically, the upstream residuals fall within the training-time coverage along the exact geometric dimensions that the model’s update rule is theoretically defined on and that we explicitly verify in our analysis.
>
> **W4 Compuataional Cost:**
>  Following previous work, we quantify computational cost using wall-clock sampling time under a fixed hardware and batch configuration. All methods are evaluated on an NVIDIA A100 with batch size = 64, using 10 sampling steps and their recommended numerical precision:
> | Method             | Average time (seconds) | Precision |
> |--------------------|------------------------|-----------|
> | ET-Flow / Refiner  | 0.03                   | FP32      |
> | MCF-L              | 0.07                   | FP32      |
> | DMT-L              | 0.04                   | FP16      |
>
> Under the same number of steps, ET-Flow (and the Refiner) achieves the lowest wall-clock cost among these baselines. This is expected because both methods share the same architecture and PC sampling strategy; they differ only in model weights and the initial input coordinates (same shape, different values). Therefore, using the number of sampling steps as a proxy for computational cost is a conservative and fair way to evaluate the Refiner.
> Exact FLOPs are not well-defined in our setting. The molecular graphs for ET-Flow, MCF, DMT, and the Refiner are dynamically constructed via a radius-based cutoff. As atom positions evolve during sampling, the neighborhood structure and number of edges (and thus the number of message-passing operations) change at every step, so the FLOPs of a single forward pass are not constant.

---

> ### Author Response · Authors · 2025-11-21
>
> **Q3 Metric and Diversity: a) Metrics:** We follow prior work in using RMSD-based recall and precision to evaluate both the accuracy and coverage of generated conformers, with formal definitions provided in Eq.(18). For recall, each ground-truth conformer queries the generated set, and we take the minimum RMSD over all generated conformers; in other words, recall directly measures whether *every* ground-truth conformer can be matched by at least one sufficiently close generated conformer. AMR summarizes the magnitude of these nearest-neighbor RMSD values, while COV reports the fraction of ground-truth conformers whose minimum RMSD is below a fixed threshold. We also report chemical validity metrics in Table~3, following previous work.
> **b).Diversity:** We include a new Appendix subsection A7 (fig8), which presents a histogram of per-conformer recall RMSD over all test molecules, showing that the recall error distribution slight lower than upstream model alone. This indicates that the Refiner preserves conformational diversity while improving accuracy.
>
> **W4 Compuataional Cost:**
>  Following previous work, we quantify computational cost using wall-clock sampling time under a fixed hardware and batch configuration. All methods are evaluated on an NVIDIA A100 with batch size = 64, using 10 sampling steps and their recommended numerical precision:
> | Method             | Average time (seconds) | Precision |
> |--------------------|------------------------|-----------|
> | ET-Flow / Refiner  | 0.03                   | FP32      |
> | MCF-L              | 0.07                   | FP32      |
> | DMT-L              | 0.04                   | FP16      |
>
> Under the same number of steps, ET-Flow (and the Refiner) achieves the lowest wall-clock cost among these baselines. This is expected because both methods share the same architecture and PC sampling strategy; they differ only in model weights and the initial input coordinates (same shape, different values). Therefore, using the number of sampling steps as a proxy for computational cost is a conservative and fair way to evaluate the Refiner.
> Exact FLOPs are not well-defined in our setting. The molecular graphs for ET-Flow, MCF, DMT, and the Refiner are dynamically constructed via a radius-based cutoff. As atom positions evolve during sampling, the neighborhood structure and number of edges (and thus the number of message-passing operations) change at every step, so the FLOPs of a single forward pass are not constant.

---

> ### Author Response · Authors · 2025-11-29
>
> **W5/W6 Multi-seed Run:** The main results are reported using a random seed of 42, which follows the official ET-Flow implementation and script. In addition, we run five experiments with different seeds (42, 0, 1) on QM9 and GEOM-DRUGS in the ET-Flow + Refiner (20 + 20) setting. The averages and standard deviations are summarized in the Table below.
>
> | Metric             |Recall-COV↑||Recall-AMR↓||Precision-COV ↑||Precision-AMR↓||
> |-|-|-|-|-|-|-|-|-
> | QM9            |Mean |Median|Mean |Median|Mean|Median| Mean |Median|
> |42(default)  | 78.66%|	89.74%|	0.0754 |	0.0273 | 76.73%|	87.50%|	0.1056	|0.0373 |
> |average  | 78.58%	|89.30%	|0.0759|	0.0268	|	76.44%|	86.00%|	0.1067|	0.0378|
> |std  | 0.08%	|0.39%|	0.0004|	0.0004	|	0.26%|	1.32%|	0.0009|	0.0005|
>
> | Metric             |Recall-COV↑||Recall-AMR↓||Precision-COV ↑||Precision-AMR↓||
> |-|-|-|-|-|-|-|-|-
> | DRUGS   |Mean |Median|Mean |Median|Mean|Median| Mean |Median|
> |42(default) |80.20%|85.71%	|0.4445	|0.4114		|73.47%	|81.25%	|0.5468	|0.4766	|
> |average	|80.05%	|85.68%	|0.4458	|0.4143		|73.39%	|81.27%	|0.5475	|0.4771	|
> |std	|0.13%	|0.03%	|0.0011	|0.0026		|0.07%	|0.04%	|0.0006	|0.0005	|
>
> We observe that the improvements are stable across seeds, with standard deviations consistently below 0.2%. This is expected because, for each test configuration, we first average over approximately 100 conformers per molecule  (average) and then over 1,000 molecules (median/mean); thus, the final AMR score is effectively an average over 100k conformers.  For this reason, prior work (including ET-Flow) typically reports single-seed results; we follow this convention in the main paper and provide multi-seed statistics here for completeness. Due to limited computational resources during the rebuttal period, we are unable to rerun all configurations with multi-seed error bars, but we will incorporate these additions and the requested annotations in the camera-ready version.
>
> **Q5/Q6 Cross-Upstream and Training Variants:**  **a).Cross-Upstream** From a computational point of view, a full cross-robustness study (training the Refiner with upstream A and refining with upstream B/C) is unfortunately beyond our reach. The large upstream models MCF-L and DMT-L are substantially heavier than ET-Flow (about 242M and 150M parameters vs. 8.3M for ET-Flow), and even for ET-Flow, a single Refiner fine-tuning run already takes about 3 days on 2×A100 GPUs. Training and evaluating separate Refiners for all pairs of upstream models would multiply this cost several times and exceed our computational budget.
> **b).Training Variants:** Similarly, the variant that sets $x_0$ to “upstream samples + slight random perturbation” during training would require repeatedly generating upstream conformers on the full training/validation split. As mentioned in Section 5.1, the train/val/test split we use is 243,473 molecules for training, 30,433 for validation, and 1,000 for testing. On the standard benchmark subset of 1,000 test molecules, each with roughly 100 conformers, ET-Flow alone already requires approximately 5 GPU hours on a single A100 （20 steps), and MCF/DMT is more expensive. Scaling this procedure to the full train+val split (approximately 270 times larger than the 1,000-molecule subset) would correspond to on the order of 75 A100 Days, which is far beyond the resources available for this project. This is an interesting direction that we would like to explore as future work if sufficient computational resources become available.

---

### Author Response · Authors · 2025-12-01
**Author Summary of Clarifications**

We sincerely appreciate the AC’s effort in reviewing the submission, and we thank the reviewers for their constructive and valuable feedback. The following provides a unified summary of our responses to all reviewers. The content is fully aligned with the point-by-point replies and is presented to help the AC assess the overall clarifications more efficiently.

**PART I Methodological Clarifications and Supporting Evidence**
1) Algorithm 1 vs. Eq. (6)–(9) by Reviewer WFCh(W1/Q1) and Xj5K(W1/Q1):
2) Training–sampling distribution difference and t-realignment by Reviewer WFCh(W2/W3/Q3) and CRoH (W1a).
3) On scope and generalizability by Reviewer CRoH(W2)

**1** The pseudocode contained a simplified line; we corrected it and showed via ablations that removing the $s(t)$ term has only a minor influence on performance.

**2** Although the training and sampling distributions differ at a global level, the per-atom geometric statistics used as model inputs (distances and directions) remain within the training range. Test-time neighbor degrees are slightly smaller, but this affects only the scale of aggregation. The expected value increases monotonically with increasing $t$; this $t$-realignment naturally aligns test inputs with the training regime. Empirical statistics match our assumptions.

**3** The Refiner is plug-and-play within the same task: it can robustly refine outputs from different upstream generators. Extending it to other tasks would require appropriate task-specific datasets and model architectures, which we view as an interesting future direction.

**PART II Additional Experimental Evidence**
1) Evaluation metrics and diversity by Reviewer WFCh(Q3)
2) Comparison to classical refinement methods by Reviewer Xj5K(W2)
3) Robustness under upstream structural bias by Reviewer CRoH(W1b) and jEgW(W3/Q1)
4) Case studies, and visualizations by Reviewer jEgW(W4/Q2)

**1** Our evaluation strictly follows previous MCG metrics by RMSD and chemical properties, and recall directly reflects conformer diversity; the results show no signs of oversmoothing.

**2** Classical refinement methods such as MMFF worsen modern generator outputs, confirming that the Refiner corrects learned generative errors that physics-based optimizers cannot

**3** Additional tests using more structurally biased upstream conformers (e.g., those generated by RDKit) show that the Refiner remains robust within the intended regime.

**4** New case studies and visualizations demonstrate substantial overall improvements, while the few mild degradations observed in edge cases stem from the inherent modeling imperfections of the deep neural networks; the Refiner’s net gains remain large and fully consistent with our robustness analysis.

---

> ### Author Response · Authors · 2025-12-01
>
> **PART III Implementation and Practical Considerations**
> 1) Multi-seed runs by Reviewer(W5/W6)
> 2) Computational Cost and Memory Usage** by Reviewer WFCh(W4/Q4) and jEgW(W2/Q3)
> 3) Step efficiency and consistency with diffusion/FM practice by reviewer  CRoH(W3)
> 4) Cross-upstream robustness and training variants by Reviewer WFCh(Q5/Q6)
> 5) Fine-tuning strategy and architectural generality Xj5K(Q2)
> 6) Reproducibility by Reviewer jEgW(W1) and Xj5K(W3)
>
> **1** Multi-seed evaluations show low variance and stable performance across runs, confirming that the Refiner’s improvements are not seed-dependent.
>
> **2** A single Refiner pass has a similar cost to an ET-Flow step, and ET-Flow is already the smallest and most computationally efficient model among recent MCG generative models.
>
> **3** The Refiner is fully consistent with standard diffusion/FM sampling paradigms and allows using fewer overall sampling steps while maintaining or improving quality.
>
> **4** Full cross-upstream training is impractical for large models, but experimental results show that the Refiner remains robust across ouputs of different generative methods
>
> **5** The Refiner is architecture-agnostic: our analysis relies on SE(3)-equivariant representations and the FM objective rather than any specific backbone, so any SE(3)-equivariant architecture should work; fine-tuning ET-Flow is computationally feasible, whereas training similarly large models from scratch is beyond our available budget.
>
> **6** Code provided in supplementary materials
>
> **PART IV Other Items**
> 1) Pipeline Illustration by Reviewer jEgW (W5)
> 2) LLM Usage Disclosure by Reviewer Xj5K(Q3)
>
> **1** We added a clear pipeline illustration as requested.
>
> **2** LLMs were used only for language polishing and grammar refinement during the writing process. All methodological formulations, theoretical motivations, and evaluations were developed entirely by the authors.
>
> **Overall summary:** All concerns raised by the reviewers can be addressed without any changes to the core methodology. The issues regarding Algorithm 1 are at the level of formulation/notation and do not affect the training objective or the achieved results. The questions on noise modeling and potential distribution mismatch are future supported by new empirical evidence (Appendix A.5/Fig 6), which shows that (i) the local geometry of each atom including distance and direction at test time lies within the training range, and (ii) the degree of neighbor can be explained by an empirically verified monotonic trend that naturally realigns the training and test distributions. Experimental concerns, including MMFF-based comparisons, robustness to upstream bias, and additional qualitative case studies, have all been carefully examined and consistently confirm the effectiveness and robustness of the Refiner. Consequently, the paper's main contributions and empirical conclusions remain unchanged. We hope that these clarifications will assist the committee in forming its final recommendation.

---

### Meta-Review · Area_Chair_6r8a · 2025-12-20

**Summary:**

This paper addresses the problem of error accumulation in denoising-based molecular conformer generation (MCG). The authors propose a Flow-Matching (FM) based Refiner that, instead of starting from pure Gaussian noise, initializes sampling from the outputs of existing ("upstream") generative models. By rescheduling the noise scale to bypass the high-noise (low-SNR) regime, the refiner corrects structural errors while maintaining diversity.

The reviewers generally agreed that the problem is well-motivated and the "plug-and-play" nature of the refiner is a significant practical strength. However, initial concerns were raised regarding mathematical inconsistencies between the algorithm and equations, the validity of the noise-calibration hypothesis, the fairness of computational cost comparisons, and robustness when facing systematic structural biases.

The authors provided a rebuttal that effectively addressed the primary technical concerns. The "typo" in Algorithm 1 was clarified with ablations showing the method's core behavior remains stable even with simplified objectives. Besides, the authors added a stress test using RDKit-generated conformers, demonstrating that the refiner can rescue extremely poor initial structures, thereby proving its robustness beyond just "small random perturbations".

My own concerns:

- A core theoretical assumption of the Refiner is that upstream geometric errors can be approximated as isotropic Gaussian noise. However, author may need to provide more evidence supporting this assumption.
- The "plug-and-play" claim is currently validated only within a specific domain. There is no evidence yet provided for how the Refiner handles transferability to domains with significantly different conformational complexity or data distributions.

**Reviewer Concerns:**

Addressed concerns:

- Mathematical Inconsistency : The authors admitted to a notation error in Algorithm 1 and provided an ablation study showing that the inclusion of the $s(t)$ term (stochasticity) has a negligible impact on performance, validating the robustness of the flow-matching framework.
- Robustness to Systematic Bias: The authors added experiments using RDKit conformers (which have systematic drift). The refiner improved RDKit's Mean AMR from 0.1728 to 0.0294, making it comparable to SOTA diffusion models.
- Comparison to Classical Methods: New results showed that physics-based optimizers like MMFF actually worsen the outputs of modern generators on the GEOM-QM9 dataset, whereas the Refiner improves them, highlighting the need for a learned approach.
- Computational Cost Fairness (WFCh, jEgW): The authors provided wall-clock timing and argued that since the Refiner uses the lightweight ET-Flow architecture, a $10+10$ step pipeline is faster and more accurate than a 100-step single-stage model.

Outstanding concerns:

- Domain Generalizability: Reviewer CROH correctly noted that the "plug-and-play" claim is currently limited to drug-like molecules in the GEOM dataset. The effectiveness on larger structures like proteins remains unproven and is acknowledged as future work.
- Anisotropic Noise Modeling: While the authors provided empirical evidence that local geometric statistics (distances/angles) at test-time fall within the training range, the theoretical assumption of isotropic Gaussian noise for the refiner's base remains a simplification of real-world error distributions.

**Reviewer Scores:**

- WFCh (possibly 4 to 6): The ablation on $s(t)$ and wall-clock timing addressed their most critical "soundness" and "fairness" blockers.
- CROH (6): The RDKit stress test directly answered their concerns regarding heuristic calibration and robustness.
- jEgW (possibly 4 to 6): The addition of Figure 2 (pipeline illustration) and the RDKit results addressed their main clarity and robustness concerns.
- Xj5K (4): The MMFF comparison resolved some weaknesses.

---

### Decision · Program_Chairs · 2026-01-26

Reject